# Carbon-Based Nanostructures as Emerging Materials for Gene Delivery Applications

**DOI:** 10.3390/pharmaceutics16020288

**Published:** 2024-02-18

**Authors:** Sara Yazdani, Mehrdad Mozaffarian, Gholamreza Pazuki, Naghmeh Hadidi, Ilia Villate-Beitia, Jon Zárate, Gustavo Puras, Jose Luis Pedraz

**Affiliations:** 1Department of Chemical Engineering, Amirkabir University of Technology, Tehran P.O. Box 15875-4413, Iran; sarayazdani@aut.ac.ir (S.Y.); ghpazuki@aut.ac.ir (G.P.); 2NanoBioCel Research Group, University of the Basque Country (UPV/EHU), Paseo de la Universidad 7, 01006 Vitoria-Gasteiz, Spain; aneilia.villate@ehu.eus (I.V.-B.); jon.zarate@ehu.eus (J.Z.); gustavo.puras@ehu.eus (G.P.); 3Department of Clinical Research and EM Microscope, Pasteur Institute of Iran (PII), Tehran P.O. Box 131694-3551, Iran; n_hadidi@pasteur.ac.ir; 4Networking Research Centre of Bioengineering, Biomaterials and Nanomedicine (CIBER-BBN), Institute of Health Carlos III, Av Monforte de Lemos 3-5, 28029 Madrid, Spain; 5Bioaraba, NanoBioCel Research Group, Calle José Achotegui s/n, 01009 Vitoria-Gasteiz, Spain

**Keywords:** carbon-based nanostructures, carbon nanotubes, carbon quantum dots, nano-diamonds, gene therapy, non-viral vectors

## Abstract

Gene therapeutics are promising for treating diseases at the genetic level, with some already validated for clinical use. Recently, nanostructures have emerged for the targeted delivery of genetic material. Nanomaterials, exhibiting advantageous properties such as a high surface-to-volume ratio, biocompatibility, facile functionalization, substantial loading capacity, and tunable physicochemical characteristics, are recognized as non-viral vectors in gene therapy applications. Despite progress, current non-viral vectors exhibit notably low gene delivery efficiency. Progress in nanotechnology is essential to overcome extracellular and intracellular barriers in gene delivery. Specific nanostructures such as carbon nanotubes (CNTs), carbon quantum dots (CQDs), nanodiamonds (NDs), and similar carbon-based structures can accommodate diverse genetic materials such as plasmid DNA (pDNA), messenger RNA (mRNA), small interference RNA (siRNA), micro RNA (miRNA), and antisense oligonucleotides (AONs). To address challenges such as high toxicity and low transfection efficiency, advancements in the features of carbon-based nanostructures (CBNs) are imperative. This overview delves into three types of CBNs employed as vectors in drug/gene delivery systems, encompassing their synthesis methods, properties, and biomedical applications. Ultimately, we present insights into the opportunities and challenges within the captivating realm of gene delivery using CBNs.

## 1. Introduction

The promising and unique gene therapy method applies genes to prohibit or meliorate any illnesses such as cancer, inherited disorders, and viral infections. In the gene treatment method, a set of certain genes that may help remedy an illness is introduced into the patient’s cells instead of applying expensive drugs or complex surgeries. Inactivating a mutated gene, substituting a mutated gene with a healthy and strong gene, and introducing safe genes into the cells to preserve them from any disorders are three approaches in gene therapy. Gene therapy is risky and has currently been studied only for special disorders that have no other efficient treatments. Genetic molecules attain the host cells’ nuclei to induce gene expression [1,2]. A comprehensive understanding of the interaction between targeting cells and the gene delivery system is a pre-prerequisite step to achieving a favorable and efficient design of the gene delivery systems. A plasmid-based gene expression system, an efficient gene that encodes a particular therapeutic protein, and a gene delivery vector are three main components of gene delivery systems [3,4]. The stability of external genetic molecules in the host’s cells is an important criterion for having an efficient and stable gene delivery system [5,6,7]. There are extracellular and intracellular barriers along the gene delivery path, which the vector should be able to overcome. Unspecific molecular interactions, endothelial barriers, and the immune response are the three main extracellular barriers. Limitations of cellular uptake, endosome–lysosome escape, intracellular trafficking, and nuclear delivery are intracellular kinds of barriers [8]. So, the correct choice of a specific vector for gene delivery is an effective parameter for the success of gene therapy. On the other hand, the vector should be able to overcome the main challenges (blood circulation, tissue pressure, endosomal escape, gene expression duration, mechanism of cellular uptake) before degradation or adsorption under biological conditions. Non-viral vectors are classified into two major groups: organic and inorganic nanostructures. Comprising carbon-based nanomaterials such as CNTs, CQDs, and fullerenes [9], NDs are a subcategory of inorganic material groups [10]. In this work, CBNs that act as non-viral vectors with high capacity in gene delivery systems are reviewed.

The description of nanostructures consists of highlighting the design and advancement of materials at the nanoscale. When these nanostructures are located in the presence of specified stimuli, they show particular responses. The modification of surface chemistry and physics via various methods helps to modulate the application of nanostructures with preferable biological properties and improve their particular operation and solubility under physiological conditions [11]. Carbon-based nanomaterials with large surface-to-volume ratios, high biocompatibility, and easy functionalization process have a high capability for participation in gene therapy. These materials can deliver different nucleic acid fragments such as plasmid DNA, siRNA, miRNA, and AONs to cell/tissue/organ targets after having been functionalized properly with biocompatible molecules [12,13,14,15].

In this review, we focus on three important nanostructures including CNTs, CQDs, and NDs. Other similar carbon-based nanostructures such as graphene oxides (GO), its reduced form (rGO), and fullerenes have shown promising results as efficient and safe genetic material delivery systems [16,17,18]. In each part, we explain the structure and properties of the nanostructure and then, introduce their synthesis methods. Finally, we describe the applications of each nanostructure in the biomedical field, especially gene delivery systems.

## 2. Carbon-Based Nanostructures

### 2.1. Carbon Nanotubes (CNTs)

#### 2.1.1. Structure and Properties of CNTs

In recent years, CNTs have attracted the attention of many researchers. CNTs were first discovered by Japanese scientist Iijima [19] in 1991, and they are one of the most applicable nanoparticles [20] in nanotechnology. CNTs are made from rolled graphite sheets produced from carbon atoms formed into hexagonal structures. The lengths of CNTs vary from lower than 100 nm to a few centimeters that span the molecular as well as macroscopic scales. CNTs are classified into two groups based on the number of graphite layers: Single-Wall Carbon Nanotubes (SWCNTs) and Multi-Wall Carbon nanotubes (MWCNTs). SWCNTs are formed from only one layer of rolled graphene with a diameter range of 0.4–3 nm. MWCNTs are made from several layers of rolled graphene sheets. The presence of more layers makes CNT’s structure more complex and subsequently more difficult to evaluate. The diameters of MWCNTs are in a range of 1–200 nm. One of the differences between SWCNTs and MWCNTs is the necessity of catalyst presence in their synthesis [21]. MWCNTs can be produced without catalysts, but SWCNTs require catalyst species such as iron-group metal, palladium, platinum [22], etc. that play an important role in their synthesis. The other difference is that MWCNT accumulates in a living body [21] more than SWCNT, which leads to SWCNT being preferred for pharmaceutical applications. In addition to the classification of CNTs according to the number of graphite layers, they can also be divided into three different types of CNTs namely, armchair, chiral, and zigzag CNTs. Each of these types of CNTs is created from a different direction of rolling graphite sheets. Chiral vector ($\vec{C}=\;n\vec{a_{1}}+m\vec{a_{2}}$) was a criterion to determine CNT’s conformation type, and is represented with two indices (n,m). Index n represents the direction of the graphite sheet, and m is related to the diameter of CNT. When m = 0 and n = m, the corresponding CNTs are called zigzag and armchair, respectively, and both structures have a mirror symmetry. The rest of CNT configuration belongs to chiral structure, which is signified by chiral angle. Some characteristics of CNT such as metallic or semi-conducting structure, diameter (d) and chiral angel (θ) can be determined by the chirality of CNTs [23]. Figure 1 represents armchair, zigzag, and chiral SWCNTs.

In the last two decades, CNTs have attracted the attention of many researchers due to their extraordinary mechanical, optical, thermal, and electrical properties. The large surface area of CNTs can conjugate to a wide variety of therapeutic molecules [24]. They can be an excellent vehicle in drug/gene delivery systems with high loading capacity to transfer drugs or genes to target cells without destroying them or causing metabolic changes in the body [25]. The bioactive drugs and genes (chemotherapeutic drugs [26] such as doxorubicin and cisplatin, antibiotics [27], nucleic acids [28], and antibodies [29]) can attach to the surface of CNT or encapsulate in the CNT structure. In this way, the toxicity and side effects of these cargoes will be reduced [30]. The high flexibility of CNTs enables them to be bent considerably without sustaining any damage and then return to the previous shape. CNTs can easily penetrate deep through membranes, and transfer cargo to target cells. Internalization of CNT to target cells is affected by the physicochemical properties of CNT or surface functionalization, cellular processes, bio-distribution, and degradation kinetics. On the other hand, the size, dimensions, and colloidal behavior of CNTs are some parameters that affect the interaction between the material that conjugates to the surface of CNT and membrane. CNT uptake at cellular level possesses different mechanisms called endocytosis, phagocytosis, pinocytosis and membrane adsorption [30,31]. Having high density-normalized Young’s moduli and good tensile strength are other properties of CNTs, which can be applied in biomedical applications [32,33]. Although CNTs possess high mechanical strength, their weight is ultra-light due to empty spaces in their structures. The chemical and thermal stability (in vacuum up to 2800 °C and in air up to 1000 °C [30]) of CNTs are considerable and are chemically unreactive [34,35,36]. Besides the advantages of CNT, there are also some disadvantages such as low solubility and toxicity that should be resolved before any biomedical applications [37,38]. To overcome these problems, CNTs should be functionalized by biocompatible and biodegradable materials to improve their dispersibility and reduce their toxicity. The functionalization of CNTs is one solution for overcoming these undesirable properties, and can be performed via two methods (covalently and non-covalently). The functionalization of CNTs is a powerful point for them to act as a smart drug/gene delivery agent, and that can stimulate multi-responsiveness. The functionalization of CNTs is explained in detail in the following section.

#### 2.1.2. Synthesis Methods of CNTs

Some prerequisites, including an active catalyst, carbon source, and sufficient energy source are needed for the synthesis of CNTs. The arc-discharge method [39,40], laser ablation [41,42], and chemical vapor deposition (CVD) [39] are significant methods for the synthesis of CNTs [43,44,45]. In the arc-discharge method, there are two graphite electrodes referred to as the anode and cathode, with the anode to be consumed so that CNT is formed on the cathode during CNT generation [46]. The source of energy in this method is provided by stable arc-discharged electricity with a voltage range of about 15–25 V [43,47]. Helium gas enters the reaction environment at low pressure (100–500 Torr, since CNT cannot be formed at Torr under 100) [48]. MWCNTs will be generated with an inner diameter of 1.3 nm and an outer diameter of about 10 nm only if unmodified graphite electrodes are used. SWCNTs with diameters in the range of 0.6–1.4 nm can be produced by using graphite electrodes with metallic catalysts such as Fe, Co, and Ni [49] at the center part of the positive electrode [43,48,50,51]. Usually, CNT synthesis through this method needs a long purification process after synthesis to remove impurities such as amorphous carbon, nontubular fullerenes, and catalyst particles. These impurities are present as part of the synthesis outcome [47,52,53]. Although some variables such as the temperature of the chamber, concentration, kind of catalyst, the presence of hydrogen, etc. affect the size and structure of synthesized CNT, this method is still chosen as the best method for producing CNT with a high degree of structural perfection [44,54]. Chaudhary et al. synthesized MWCNT by using the arc-discharge method with hydrocarbon as a feedstock at 100, 300, and 500 Torr pressures and three different arc currents [55]. Arora et al. investigated the effect of temperature on the synthesis of MWCNT from a carbon block as a precursor via the arc-discharge technique in an argon atmosphere. Also, they shifted the arc current from 25 to 40 A to study the morphological changes during the synthesis process and concluded that carbon black can convert to MWCNT even at low levels of current and when the arc temperature is kept constant for longer periods [56]. Singh and coworkers produced MWCNT via the AC arc-discharge method from pure graphite electrodes of different shapes and reported the range of nanotubes’ length to be between 231 and 561 nm, and its range diameter between 14 and 55 nm [57]. Sari et al. were able to produce MWCNT via the arc-discharge method and investigated the effect of liquid medium on the growth, size, and quality of MWCNT structure. They reported that NaCl concentration has a direct relationship to the length of generated nanotube, namely, as the NaCl concentration increases, the length of produced nanotube changes from a few micrometers to more than 150 Â µm [58].

Applying the laser ablation method for CNT synthesis was first reported by Guo and coworkers in 1995 [41]. The laser ablation method is a physical vapor deposition method with a slightly more complicated system than the arc-discharge method. Solid graphite irradiated by a laser source is introduced as the carbon source in this method and vaporized into vapor carbon atoms at 800–1500 °C. High-intensity light from a CO_2_ laser or pulsed laser source such as neodymium-doped yttrium aluminum garnet (Nd:YAG) is applied as the energy source [48]. A constant flow rate of argon or nitrogen gas enters the system housing at 500 Torr and graphite under high intense laser pulses explodes, and finally, CNT will be formed on water-cooled copper [43,47]. The laser ablation method is a good method for large-scale production of SWCNT (1–10 gr), but it consumes a lot of energy in addition to requiring different pieces of equipment [59]. So, from an economic point of view, it is not suitable. SWCNT synthesis through this method will produce diameters between 1 and 2 nm. MWCNT can also be produced through this method but requires high expenses [43]. Synthesizing SWCNT by this method will provide higher quality and yield than those obtained by the arc discharge method. Also, a narrower SWCNT distribution can be obtained through laser ablation [47]. However, both methods (arc discharge and laser ablation) consume a lot of energy to convert carbon atoms into a CNT structure [47]. The advantages of these methods are the capability to control some vital parameters such as purity of produced CNT, density, length and diameter of CNTs [43]. Chrzanowska and co-workers investigated the effect of laser wavelength on properties of SWCNT and production yield [60]. They concluded that laser fluence has a stronger effect on the properties of synthesized nanotubes than infrared laser radiation [60]. Das et al. tried to optimize the arc-discharge and laser ablation methods for CNT synthesis [61]. Their attempts produced a CNT yield with the desired diameter and chirality [61]. Alamro et al. functionalized CNT with different amounts of silver nanoparticles through a laser-ablation-assisted method to be used as an excellent adsorption material against naphthalene [62]. They concluded that loading silver nanoparticles on the surface of CNT provides an option for water treatment applications [62]. Zhang and coworkers synthesized SWCNT using the laser ablation method at room temperature. Co and Ni as a catalyst and graphite powder were transformed into a target plate by heat treatment [63]. Table 1 indicates some research related to CNTs synthesized via the laser ablation method.

Chemical vapor deposition (CVD) is another method for the synthesis of carbon nanotubes. Here, the carbon source is one such as ethanol, acetylene, propylene, and methane that enters the processor in the presence of a metal catalyst (Ni, Fe, Co, etc.), and is heated to 1000 °C [44,71,72]. The diameter of catalyst particles and the length of the reaction time affect the diameter of MWCNT produced by this method [73]. The CVD method can be used for large-scale CNT production. Good alignment, control over diameter and its wall number as well as control of nanotube growth are some of the powerful and positive points of this method [72]. This method can be divided into seven different techniques (hot film CVD, plasma-enhanced CVD (PECVD), radio frequency PECVD, microwave PECVD, water-assisted CVD, oxygen-/carbon dioxide-assisted CVD, and CVD with organo-metallic precursors) [74]. Szymanski et al. developed a new approach for CNT synthesis in reactors operating at atmospheric pressure [75]. Zhang et al. focused on upgrading the controlled synthesis of ultra-long CNT with perfect structures and excellent properties [76]. Duc Vu Quyen et al. synthesized CNT from liquefied petroleum gas on a Fe_2_O_3_/Al_2_O_3_ pre-catalyst via a CVD method without hydrogen [77]. Yang and coworkers synthesized CNT-microfibrous composites via the thermal CVD method and studied the effect of different parameters such as deposition time and temperature, acid pretreatment, and gas flow rate on the properties and structure of the obtained composites [78]. Ibrahimov et al. synthesized CNT via CVD technology and pyrolysis of natural gas in the presence of iron at the nano-scale [79]. Arunkumar and coworkers produced large-scale MWCNTs via the thermal catalytic CVD method by applying acetylene and Fe/MgO as gas and catalyst, respectively. The characterization results indicated that the particle size of MWCNT was in a range of 20–30 nm [80]. Das et al. reported that the size of catalyst nanoparticles plays an important role in the diameter size of MWCNTs when synthesized through microwave plasma-CVD at low-temperatures [81]. Zhang et al. synthesized and then characterized MWCNTs via microwave-assisted CVD and also studied various parameters (catalyst, carbon source, the renewable carbon substrate, and the reaction’s temperature) [82]. The results showed that 600 °C was the optimum temperature for producing MWCNTs. The length of MWCNTs was between 2600 and 3200 nm and the MWCNT’s diameter was 50 nm [82]. Smagulova and coworkers synthesized CNT from high-density polyethylene waste in a one-step CVD process [83]. The results of the physicochemical analysis confirmed the absence of turbostratic carbon in the final products [83]. Tang and coworkers nominated some prominent methods in progressing the quality of CVD-grown 2D components [84]. Haque et al. synthesized MWCNT via CVD process and converted it to a diamond nanostructure by nanosecond pulsed laser melting procedure at room temperature and without using any catalyst [85]. Table 2 indicates some studies regarding CNTs synthesized via CVD method.

CNT purification is the next step after the CNT synthesis process. In this way, unfavorable entities such as carbon nanoparticles, residual catalysts, and other graphitic impurities are separated [97]. Choosing an efficient and gentle purification method is necessary to prevent the destruction of the CNT’s structure. The purification process can be done by both chemical and physical techniques [97]. Most of the purification methods are applied simultaneously to eliminate the impurities and improve the purification process. Liquid and gas phases are part of the chemical purification category and filtration, sonication, and chromatography are a subset of the physical purification category. The oxidation of synthesized CNTs is more applicable via the chemical purification process at both dry and wet conditions. Oxidation at dry conditions is processed by air and oxygen, and highly concentrated acidic solutions or strong oxidants are chosen for the oxidation process at wet conditions. The simplicity and high yield of chemical oxidation have attracted extensive attention in liquid-phase technology. Concentrated acid can remove the metal particles and carbon impurities from CNT. The oxidation period and reflux temperature are two parameters for eliminating catalyst particles [97,98]. The physical purification process is applied to reduce the damage via non-conventional methods. The ultra-sonication procedures are carried out via the segregation of particles. High energy in the presence of solvents such as o-dichlorobenzene and dichloromethane is applied for purifying CNT by ultra-sonication methods. During these purification procedures, the molecules can interact with CNTs and create solubilization. Filtration is another method of physical purification that depends on different properties such as size (length and diameter) and particle separation. On the other hand, chromatography is used for purification and cutting the length of CNTs. High-performance liquid chromatography (HPLC) and size exclusion chromatography (SEC) are two popular methods for length separation of CNTs. Chemical purification is superior compared to physical purification due to its moderate conditions, which purifies CNT weakly. However, the positive characteristic of physical separation is the capability to reject impurities during production [97,99].

#### 2.1.3. Functionalizing CNTs

As mentioned in the previous section, CNTs have some unwanted properties that create problems for biomedical applications. Two of the most significant problems of CNTs are high toxicity and low solubility. CNT’s solubility is very weak in most solvents (both organic and inorganic) due to possessing highly hydrophobic surface structures. The solubility factor of CNTs in aqueous solutions should be seriously improved before being applied in biological systems [74]. Furthermore, the toxicity nature of CNTs should also be eliminated. Thereupon, the surface of CNTs should be modified by materials that will improve their physicochemical properties. So, functionalizing CNTs with organic and inorganic materials will increase their dispersion and biocompatibility and reduce their toxicity [100]. Generally, there are two methods of functionalizing CNTs: covalent and non-covalent, which are addressed in the next section.

Covalent functionalization of CNTs: This method has been widely used for increasing dispersion and easy conjugation to bio-molecules. In this method, the functional groups attach to the surface of CNT with covalent bonds [74]. The hybridization structure of CNT changes from sp^2^ to sp^3^ state by molecules with high chemical reactivity used for this process [101]. Fluorination of CNTs was the first investigation of covalent-functionalization of CNTs through this method [101]. C-F bonds on sidewalls of fluorinated CNT are weak and, therefore, create sites for additional functionalization such as amino and hydroxy groups. In addition to the fluorination of CNTs, other methods such as carbene and nitrene addition, cycloaddition, chlorination, and bromination can also be used [101]. Le et al. [102] synthesized poly(ionic liquid)s from poly (styrene-alt-maleic anhydride) (PSM) and furfuryl amine, which is called PSMF-IL. In the next step, they conjugated PSMF-IL to the surface of MWCNT to produce PSMF-IL-MWCNT nanomaterial. They observed four peaks in C1s of MWCNT, which related to C-C, C-O, C=O, and π-π* transition level. Also, they observed that the intensity of O1s increased considerably after functionalizing MWCNT, and two new peaks of N1s appeared at PSMF-IL-MWCNT, which relate to C-N and C=N bonds. He et al. reported an impressive method for covalent functionalization of CNT. This method includes decreasing carbon nanomaterials with sodium naphthalide, and then, adding diaryliodonium salts, arene, and heteroarene iodonium salts [103]. Guzman and coworkers functionalized CNT-COOH covalently with insulin as a protein model [104]. In this approach, candida antarctica lipase was used as an enzyme that can help the conjugation of carboxylic groups to hydroxyl groups [104]. Table 3 indicates some studies regarding covalent-functionalized CNTs.

Non-covalent Functionalization of CNTs: In this method, unlike the covalent approach, it is preferred that the structure of the nanotube does not change due to functionalization. The non-covalent functionalization of CNT generally includes three steps: ultrasonication of CNTs in solution, centrifugation, and finally filtration. These steps are carried out in a room-temperature environment. In non-covalent functionalization, a wide variety of compounds such as bio-molecules, polymers (such as polyethylene glycol and its derivatives), and surfactants (like sodium dodecyl sulphate) and polynuclear aromatic compounds can be attached to the surface of CNT through physical weak interaction. On the other hand, different forces such as π-π stacking interactions, van der Waals forces, hydrogen bonds, and electrostatic forces between CNT and molecules create and help to modify the initial structure of CNTs [74]. Siu et al. functionalized SWCNT non-covalently with succinate polyethyleneimine, and applied it as a carrier for siRNA delivery into melanoma [115]. Wang and coworkers functionalized MWCNT with porphyrin-Sn networks for increasing protein adsorption [116]. The final product was characterized, and the results showed that the functionalized CNT was capable of adsorbing high levels of proteins (cytochrome, lysozyme, bovine hemoglobin, and bovine serum albumin) [116]. Yazdani et al. functionalized SWCNT non-covalently with DSPE-PEG-COOH and then applied it as a carrier for amphotericin B delivery, and also investigated the capability of functionalized CNT in gene material delivery [117]. Yazdani et al. also studied the stability of the overall amphotericin B delivery system by the Flory–Huggins model [118]. Hadidi et al. functionalized SWCNT non-covalently with an amine derivative of phospholipid polyethylene glycol to decrease the intrinsic toxicity level of pure CNT [119]. The toxicity cells were carried out on the lung, liver, and ovarian cancer cell lines and indicated the CNT functionalizing process has an efficient role in applying the functionalized CNT in photothermal therapy [119]. Table 4 indicates some studies regarding non-covalent functionalized CNTs.

#### 2.1.4. Application of CNTs in Gene Therapy

CNTs provide numerous applications in various fields due to their inherent physicochemical properties such as distinctive length-to-diameter ratio, biocompatibility, nanotube surface area, and easy chemical functionalization by different materials. The pure and pristine CNTs have a hydrophobic structure and disperse with difficulty in aqueous media. Functionalizing CNTs (covalently or non-covalently) helps to increase their solubility and improve biocompatibility by modifying their surface with different molecules and creating functional groups such as carboxylic acid or amine groups, which makes them ideal for delivering nucleic acid, plasmid DNA, siRNA, AONs, and aptamers [127]. Related to this, shown in Table 5, there are a lot of works where researchers elaborate on functionalized CNTs for genetic material delivery. The following are the main examples for each type of nucleic acid. Pantarotto et al. [128] functionalized CNTs with ammonium and used for plasmid deoxyribonucleic acid (pDNA) delivery. They observed the functionalized CNTs have a low cytotoxicity and penetrate the cell easily [128]. Krajcik et al. [129] functionalized SWCNT chemically with hexamethylenediamine (HMDA) and poly (diallyldimethylammonium) chloride (PDDA) to obtain a nanocarrier to conjugate to siRNA by electrostatic interactions. They concluded that PDDA-SWNT is an effective carrier system for intracellular delivery of siRNA [129]. In the work of Siu et al. [130], three different combinations were made from a lipopolymer (DSPE-PEG-COOH) and polyethyleneimine (PEI) at different ratios and called DGI. Next, they used three DGIs as a surfactant for functionalized carbon nanotubes. Finally, they attached the functionalized CNT to siRNA and characterized it at in vitro and in vivo scales. They concluded that the combination of DGI with low amounts of PEI acted effectively for complex siRNA-CNT, and the cytotoxicity of three different DGI combinations was low in in vitro experiments [130]. Mohammadi et al. [131] synthesized carboxylated SWCNT and then conjugated it to polyethylenimine-piperazine to be used as a vector to fabricate vector-aptamer for targeted delivery of siRNA. Massoti et al. [132] functionalized CNT with two polymers (polyethyleneimine (PEI) and polyamidoamine dendrimer (PAMAM)) and conjugated it to miR-503 oligonucleotides. They reported that the toxicity of CNT was reduced after coating it with the two polymers, and it also helped the efficiency of miR-503 oligonucleotide delivery to endothelial cells [132]. They concluded that these formulations increase the stability of miRNA oligonucleotides in serum [132]. Gu et al. [133] functionalized MWCNT with polyethylene glycol (MWCNT-PEG) and conjugated it to anti-PSMA aptamer (MWCNT-PEG-Ap) to target prostate cancer cells and investigate (MWCNT-PEG-Ap) at in vitro and in vivo scales. They observed the ultrasound signal in MWCNT-PEG-Ap was stronger than that in MWCNT-PEG and no ultrasound signal was detected in the heart. They concluded this formulation (MWCNT-PEG-Ap) had a good effect at in vivo scale [133]. Table 5 shows the exhaustive overview of the functionalized CNTs to deliver nucleic acid in both in vivo and in vitro cases. The studies in the following table are categorized according to the type of nucleic acids delivered (pDNA, siRNA and miRNA, AONs and aptamers). Chen and coworkers [134] modified the SWCNTs with amylose derivatives containing poly(L-lysine) dendrons (ADP@SWNT). They studied some parameters such as aqueous dispersion stability, cytotoxicity, gene transfection efficiency, and photothermal effect of the complex both in vitro and in vivo. Figure 2 illustrates the tumor-inhibiting effects of ADP@SWNT/TNFα and ADP@SWNT/TNFα combined with laser irradiation in a nude mouse model of human colorectal cancer [134]. In Figure 2a, the absence of tumor necrosis was evident in the ADP group. Similarly, the ADP@SWNT/TNFα group displayed no significant damage, albeit with observable irregularities in shape. In contrast, the ADP@SWNT/TNFα+laser irradiation group exhibited a pronounced level of injury, coupled with a reduction in tumor size due to thermal combustion on the skin. These findings underscore the efficacy of the near-infrared (NIR)-triggered photothermal effect of ADP@SWNT/TNFα in selectively damaging tumor cells, thereby suggesting its potential utility in tumor therapy [134]. As shown in Figure 2b, initial tumor sizes were comparable across the three groups. Subsequently, the ADP group exhibited substantial tumor development, distinguishing it from the other two groups after 5 days. By day 19, the ADP group’s tumor reached 4 cm^3^, while the ADP@SWNT/TNFα group’s tumor was 1.8 cm^3^. The observed anti-tumor effect in the ADP@SWNT/TNFα group resulted from TNFα delivery via the complex into the tumor. TNFα encoded by plasmid DNA inhibited tumor growth, and the SWNTs’ enhanced membrane penetrability increased damage to tumor cells. For the ADP@SWNT/TNFα+irradiation group, laser irradiation at 808 nm on day 19 yielded a tumor size similar to the ADP@SWNT/TNFα alone group for the initial 18 days. Remarkably, tumor size in the ADP@SWNT/TNFα+irradiation group decreased by over 50% post-irradiation, indicating effective and rapid triggering of ADP@SWNT/TNFα’s light–heat conversion characteristics by near-infrared (NIR) irradiation, leading to tumor damage. This underscores the potential of NIR irradiation as a mechanism for efficiently harnessing the light-heat conversion properties of ADP@SWNT/TNFα in tumor therapy [134]. As illustrated in Figure 2c, an irregular structural pattern was evident in the colorectal cancer tissue of the ADP group. Nevertheless, the cancer cells exhibited completeness and generally maintained a favorable status. In the case of the ADP@SWNT/TNFα group, partial impairment of the tumor structure was observed, accompanied by observable cell necrosis. Furthermore, in the context of ADP@SWNT/TNFα+ laser irradiation, a heightened cytotoxic effect was evident, manifesting as pronounced structural damage to the tumor and induction of cell necrosis [134].

Despite these promising results obtained at the preclinical level, actually, there are no clinical assays on the way to evaluate the performance of CNTs in the gene therapy field, although these structures are under phase I clinical evaluation for drug delivery to face non-microcytic and advanced lung cancer.

### 2.2. Carbon Quantum Dots (CQDs)

#### 2.2.1. Structure and Properties of CQDs

In recent years, small carbon nanoparticles called CQDs have become a new group of carbon nanoparticles with a zero-dimensional structure, less than 10 nm in size, and relatively strong fluorescence characteristics [158]. This type of fluorescent carbon-based nanomaterial was first discovered by Xu et al. [159] in 2004 during the purification of SWCNTs. Unlike CNTs, CQDs have better water solubility and also, unlike nanodiamonds, CQDs can be prepared and separated easily. Graphene quantum dots (GQDs), carbon nanodots (CNDs), and polymer dots (PDs) are three sub-categories of CQDs [160]. The difference between these three sub-categories is in their internal structure. Their structures are monodisperse spherical based on the carbon atom and chemical groups including oxygen located on their surface. Their appealing properties such as high chemical stability, low toxicity, and good conductivity have attracted the interest of many researchers. In contrast to carbon materials that have poor solubility in most of the aqueous solvents, as well as weak fluorescence, the carboxyl moieties on the surface of CQDs tend to have good solubility, strong photoluminescence emission and optical properties [161]. Because of this unique and superior property, CQDs have been applied in biomedicine (bioimaging, biosensor) applications. Also, they serve as an efficient carrier in biomolecule/drug delivery operations [162,163]. Excellent optical properties and good chemical and photochemical stability are superior properties of CQDs when compared with semiconductor quantum dots, which make CQDs desirable to apply in bio-imaging. The non-toxicity and environmentally friendly characteristics are premium features of carbon, which make CQDs applicable in biological systems at both in vitro and in vivo scales [164]. Hsu et al. synthesized highly water-soluble and biocompatible carbon nanodots with an average diameter of 3.4 ± 0.8 nm and a quantum yield of 4.3% [165]. The results of the prothrombin time assay of plasma confirmed the high biocompatibility of CQDs, and the inhibition yield for MCF-7 and MDA-MB-231 cancer cells were up to 80% and 82%, respectively, which confirmed the high capability of CQDs as cancer inhibitors [165]. Huang et al. also studied the efficacy of the injection routes on the distribution, clearance, and tumor uptake of CQDs [166]. They concluded that CQDs are rapidly and impressively repelled from the body when the injection routes were intravenous, intramuscular and subcutaneous [166]. Andrius et al. investigated the potential of CQDs coated with a silver shell as a sensitizer for both radiotherapy and phototherapy in cancer cells at an in vitro scale [167]. They concluded that cell viability reduced after radiation exposure [167]. Furthermore, CQDs can be functionalized with biocompatible bio-molecules and applied as a nanocarrier for drug/gene delivery systems [168]. Wang et al. modified the structure of CQDs with hyaluronic acid and carboxymethyl chitosan through a one-step hydrothermal method and applied it for doxorubicin delivery [169]. Doxorubicin release data were pH-dependent and the in vivo results showed a considerable improvement in cancer tumor therapy [169]. Wang et al. synthesized highly fluorescent CQDs through a hydrothermal method that included citric acid and O-phenylenediamine, for loading doxorubicin drug based on donor-quenched nano-surface energy transfer [170]. The quantum yield of synthesized CQDs was 46% and showed an excellent cytocompatibility [170].

#### 2.2.2. Methods of Synthesizing CQDs

In recent years, carbon sources and reaction processes have become the two important parameters for choosing the CQDs’ synthesizing methods [171,172]. Generally, CQDs’s synthesizing methods can be classified into two groups: top–down and bottom–up (Figure 3) [173]. Generally, GQDs are two-dimensional nanoparticles, that are produced by the top–down method. In this approach, macroscopic carbon structures with distinct graphene networks such as carbon black, activated carbon, CNTs, and graphite powder are scratched and cut. Long processing times, rough reaction conditions, and expensive ingredients and equipment are negative points of top–down method [174,175]. This approach is appropriate for large-scale production. In contrast, CNDs and PDs are three-dimensional nanoparticles with spherical cores, which are obtained by bottom–up strategies. Polymerization and carbonization of molecular precursors such as citric acid, glucose, and sucrose via hydrothermal and solvothermal, plasma and microwave syntheses, and thermal decomposition generate CNDs and PDs with low defects and high controllability [176]. The optimization of synthesized CQDs can be done during preparation or post-treatment for selected applications. Three problems should be during CQD preparation to produce CQDs with great properties for application in the biomedical field. One of the problems is size control and uniformity that can be optimized through post-treatments such as centrifugation and dialysis. The second problem is related to surface properties (an important factor for the solubility of CQDs) that can be controlled either during preparation or post-treatment. The third problem is carbonaceous aggregation during carbonization which can be prevented by electrochemical synthesis and chemistry solution methods [161]. In the following, two original methods for CQD synthesis are discussed in full detail.

##### Top–Down Methods

In the top–down approach, bulk materials such as CNTs, carbon soot, NDs, activated carbon, and graphite oxide are finely stripped, and cut by physical and chemical processes into tiny and delicate materials (especially CQDs) that are in the nano-scale range [177]. The physical and chemical techniques applied in this approach include laser ablation, ultrasonic synthesis, electrochemical carbonization, and arc-discharge methods.

Laser Ablation Method: In this method, inorganic nanoparticles are produced from a solid substrate; in other words, the laser beam delivers high energy to an object’s surface and this material is either ablated or removed, and the nanoscale dimension is documented [162]. Laser ablation is a beneficial method for making CQDs with a closed-size distribution, great water solubility, and fluorescence [178]. The range of the fluorescence quantum yields of CQDs is from 4% to 10% at 400 nm [179]. This technology is a speedy and impressive method and is capable of adjusting CQD’s surface. However, the particle size of produced CQDs is not controllable in this method, which makes for low quantum yields [179]. One of the first published reports that was based on this method was authored by Y-P Sun and colleagues in 2006 [180]. By using high-temperature treatment of a graphite powder/cement mixture, they prepared a carbon target and then applied laser ablation. A high temperature of around 900 °C was necessary for producing carbon nanoparticles. They observed that the carbon dots produced by laser ablation did not have fluorescence properties before and after purification and treatment with a strong acid. So, they functionalized the surface of carbon dots (CDs) with polyethylene glycol and observed that the functionalizing process plays an important role in creating the desired photoluminescent properties. They concluded that the particles’ surface, especially the “surface defects” that should absorb light at a specific wavelength, has a significant role in the fluorescence properties of CDs. Other researchers [180] also used propinylethylenimine-co-ethylenimine (PPEI-EI) polymer for passivation of CDs’ surface. They concluded that surface passivation has a considerable effect on the proliferation of CD studies that enables carbon dots to have a hydrophilic structure and be soluble in aqueous solvents, which can be useful for biological applications such as bio-imaging. Carbon-based nanomaterials that can be utilized as a building unit for producing CDs include organic solvents, various saccharides, amino acids, proteins, and others. According to a thermodynamic point of view, the formation of graphitic nanoparticles is an important and essential feature in carbon dot preparation methods [181]. Kaczmarek et al. [182] synthesized fluorescent CQDs through nanosecond laser ablation of graphite in polyethyleneimine and ethylenediamine. The process products were detached by dialysis. The optical results indicated that the source of luminescence of produced nanostructure is fluorescent particles or quasi-molecular fluorophores [182]. Ren and coworkers [183] built N-doped micro-pore CQDs with a quantum yield of 32.4% and a fluorescence lifespan of 6.56 ns. The final products showed a stable and independent status at different conditions (various pH, temperature, salt concentration, etc.), and can be a source for biomedical and engineering applications [183]. Cui et al. [184] fabricated CQDs from cheap carbon fabrics by using the laser ablation method. The quantum yield of produced CQDs was 35.4%, and was applied in biomedical imaging. CQDs give out more photoluminescence based on the appropriate solvent and molecular precursor [184].

Ultrasonic Method: In this approach, ultrasound application creates high and low-pressure waves and produces small vacuum bubbles under these conditions. This phenomenon known as cavitation, creates strong hydrodynamic shear forces and de-agglomeration. Carbon materials with a macroscopic scale can be converted into nanoscale CQDs under the impact of ultrasonic waves [185]. This technique is very simple and does not require a lot of equipment [186]. Park et al. used waste food as the carbon source for the synthesis of CQDs at room temperature through ultrasonic treatment [187]. Results of HRTEM and AFM images of the final product showed a uniform spherical shape with an average size of 4.6 nm [187]. One stable solution of CQDs was produced by He et al. [188] from activated carbon by hydrogen peroxide (H_2_O_2_)-assisted ultrasonic treatment. The stability of the produced CQDs was high even after 6 months [188]. Qi and coworkers [189] used L-glutamic acid as a molecular precursor for synthesizing N-doped CQDs through an ultrasonic-assisted hydrothermal procedure at two different temperatures. The results indicated that one of the obtained N-doped CQDs was full of pure amorphous carbons and the second was a mixture of pure amorphous carbons and pyroglutamic acid. The quantum yield of produced CQDs was 40.5%, and the results confirmed the potential of the final products for application in bio-imaging [189]. Dang et al. [190] fabricated large-scale white fluorescent CQDs from oligomer polyamide resin through an ultrasonic process at room temperature, and used them as ink to make luminescent patterns. The quantum yield of produced CQDs was 28.3%, and the generated CQDs’s sizes were in the range of 2–4 nm, and they showed excellent dispersion and low crystallinity [191]. Li and coworkers [192] used starch soluble as a molecular precursor to synthesize blue-emitting CQDs by an ultrasonic technique. In the next step, the obtained CQDs were functionalized with *γ*-methacryloxy propyl trimethoxyl silane, and then were mixed with silicon rubber to make CQDs-silicon rubber composite [192]. The size distribution of the original product was in the range 1–4 nm, and it displayed a remarkable transparency and excellent thermal stability [192]. Xu et al. [193] made nitrogen-doped multicolor CDs from kiwifruit juice via ultrasonic technique by including various additives such as acetone, ethanol, and ethylenediamine. Results indicated that the original output had a high potential in food chemistry and anti-counterfeit applications [193].

Electrochemical Carbonization: Electrochemical carbonization is a powerful technique for synthesizing CQDs from bulk carbon materials under normal temperature and pressure conditions [194]. Specifically, graphite rods are applied as cathodes and anodes in an electrochemical setup that includes NaOH/Ethanol as the electrolyte solution. CDs produced from this method exhibit different colors and do not have uniform spherical morphologies [181]. Deng and co-workers [195] utilized low-molecular-weight alcohols for generating CQDs under basic conditions through electrochemical carbonization. The TEM images of propanol CD (PC-dots), butanol CD (BC-dots), ethylene glycol CD (EGC-dots), and glycerine CD (GC-dots) show that their diameters are in the range of 3–6 nm and have an amorphous structure. Four solutions (PC-dots, BC-dots, EGC-dots, and GC-dots) represent strong blue fluorescence under their maximum excitation [195]. Ahirwar and coworkers [196] fabricated GQDs and graphene oxide quantum dots via the electrochemical exfoliation method. Two graphite rods played the electrode roles and the electrolyte solutions were a mixture of citric acid and alkali hydroxide in water [196]. The size of the obtained GQDs was in the range of 2–3 nm and a blue to green luminescence at 365 nm was clear for both GQDs and graphene oxide quantum dots [196]. Zhou et al. [197] synthesized mass production of nitrogen-doped CQDs via electrochemical technology, which included prebaked carbon and ammonium bicarbonate as an anode and electrolyte, respectively. They investigated the ability of the original product as an inhibitor to decrease corrosion of copper and observed the corrosion inhibition yield to be 96.32% [197]. Zhou and coworkers [198] fabricated GQDs in a two-electrode cell via an electrochemical exfoliation process. Carbon fiber bundles were used as the anode, and Ti mesh was applied as the cathode with H_2_SO_4_ as the electrolyte source [198]. Danial et al. [199] studied the electrochemical exfoliation technique for making a GQD suspension. The citric acid and sodium hydroxide mixture formed the electrolyte solution. Pristine graphite rods were used as electrodes without any calcination, to prevent high energy consumption. They investigated the effects of different parameters such as reaction time and voltage on the produced GQDs [199]. The results showed that, as the sodium hydroxide concentration, process time, and voltage were increased, the GQDs’s exfoliation also increased [199].

Arc-discharge Method: the first strategy for producing CQDs was the arc-discharge method. As mentioned above, Xu et al. [159] discovered fluorescent carbon-based nanomaterials during the purification of SWCNTs that was performed based on the reaction between nitric acid and the arc-discharged soot [159]. In fact, they were able to produce three kinds of carbon-based nanomaterials that were different in molecular mass and fluorescence properties. The results of Xu’s research showed that the surface of produced CQDs can be attached to carboxyl groups to improve water solubility, which would facilitate biological applications. However, due to the large particle size distribution, the specific surface area of CQDs decreased and it affected the active reaction sites in the electro-catalytic process. Due to problems such as quantum yields, size of CQDs, and uniformity of CQDs, this approach was not accepted. Arora and Sharma [200] reported in 2014 that the carbon atoms decomposed from the bulk carbon precursors in anodic electrode, that were obtained in a sealed reactor in the presence of produced gas plasma, can be reorganized by the arc-discharge method. High-energy plasma can be produced at high temperatures (~4000 K) in the reactor. In the cathode part, CQDs can be generated from carbon vapor assembly. Biazar et al. [201] synthesized high-purity CQDs and CQDs/TiO_2_ composite by the arc-discharge approach between graphite electrodes. The main reasons for using TiO_2_ in CQDs/TiO_2_ composite were to help increase the photocatalytic activity of CQDs and also, decrease the combination rate of the electron-hole. The CQDs/TiO_2_ composite is stronger than TiO_2_. Chao-Mujica and coworkers [202] used the submerged arc-discharge in water for producing fluorescent CQDs and studied different parameters such as natural phase separation, simplicity and scalability. The size distribution of obtained CQDs was between 1 and 5 nm.

##### Bottom-Up Methods

In this approach, molecular precursors change to carbon quantum dots through different methods. In fact, the high energy of microwave, hydrothermal, and ultrasound generate nanoscale-size CQDs from small organic molecules [177]. The produced CQDs have high quantum yields and excellent optical properties. The advantages of these methods are simple operating processes and controllable reaction conditions. Also, the raw materials are inexpensive [177]. Hydrothermal treatment, electrochemical methods, microwave synthesis, and thermal decomposition are subcategories of bottom–up methods.

Hydrothermal Method: In this approach, small organic molecules such as glucose, chitosan [203], citric acid [204], fructose, amino acids, and other natural products produce CQDs at high temperatures and pressures [177]. This method is cheap, non-toxic, and environmentally friendly. Mehta et al. [164] fabricated highly fluorescent CQDs from *Saccharum officinarum* juice through a single-step hydrothermal operation at constant temperature (T = 120 °C). After heating, extraction, and centrifugation, highly blue fluorescent CQDs with uniform size (~3 nm) were produced. Bhunia et al. [205] synthesized two kinds of CQDs. The first one was hydrophobic CQDs generated from mixing different amounts of carbohydrates with actadecylamine and octadecene. The hydrophilic type of CQDs was generated from heating an aqueous solution of carbohydrates within a wide pH range. If the aqueous solution of carbohydrates is mixed with concentrated phosphoric acid, hydrophilic CQDs with red and yellow emissions will be generated. Yang et al. [203] synthesized fluorescent CQDs by hydrothermal carbonization of chitosan that was functionalized with amine. These types of CQDs can be applied as new bio-imaging agents. Zhu et al. [206] used glucose and potassium phosphate as the initial materials for producing CQDs by a simple, effective, and one-step hydrothermal process. The produced CQDs were applied as an electroluminescence sensor. The amount of potassium phosphate affects the fluorescent colors of generated CQDs, and they investigated the optical and electronic properties of produced CQDs by density functional theory [206]. Nammahachak et al. [207] controlled the average size of CQDs by applying different filling volume of sucrose solutions in the hydrothermal reactor and reported that the filling volume and average size of CQDs have an inverse correlation. Kumar and coworkers [208] synthesized CQDs composed of carbon, oxygen, and nitrogen from cow milk as a starting material via hydrothermal treatment. The shape of produced CQDs was almost circular with average size of 7 nm. They concluded this technology is simple, and safe for the environment and does not need special chemical ingredients or equipment [208].

Electrochemical Method: The electrochemical strategy is a convenient and straightforward path for generating CQDs at ambient temperature and pressure [209]. Most of the synthesized CQDs produced through this method possess high stability and uniform size distribution and are applied in bio-imaging and sensor applications [209]. Hou et al. [210] synthesized CQDs with size 2.4 nm from sodium citrate and urea in de-ionized water via this strategy and reported that it can be applied as an Hg^2+^ detector. Tian and colleagues [211] reported that nitrate solution is the best choice for producing CQDs via the electrochemical exfoliation method due to the high similarity of photoluminescence and average size of 3.5 nm. The XPS and FTIR results showed some simple functional groups on the surface of obtained CQDs during electrochemical exfoliation, and the quantum yield of produced CQDs was 5.6% [211]. Niu and coworkers [212] synthesized green-fluorescent N-doped CQDs via a simple electrochemical strategy, and the quantum yield was 30.6%. Pyrocatechol and ethylenediamine were used as precursors and electrolytes, respectively. They concluded the original output of the process has a high potential in both in vivo and in vitro stages of clinical diagnosis applications [212]. Bortolami et al. [213] reported the synthesis of L-proline-based chiral CDs and ethanol-derived L-proline-based chiral CDs via electrochemical method. They optimized the electrochemical setup and studied the effect of reaction conditions on the final products, photoluminescence, and catalytic activity [213].

Microwave Irradiation: One of the fastest and lowest-cost methods for producing CQDs is microwave irradiation technology [172]. Originally, Xu and colleagues [214] utilized a mixture of calcium citrate and urea solution to generate CQDs through this method. A high luminescence was emitted (both in liquid and solid phases) by the product. Some researchers have reported [214] that green luminescent CQDs can be produced through microwave irradiation by using sucrose as the carbon source and diethylene glycol (DEG) as the reaction media. The produced CQDs possessed good dispersibility in water and their appearances were transparent. Also, this product (DEG-CQDs) was observed to have low cytotoxicity level, and could hopefully be applied for bio-imaging [172]. Liu et al. [215] generated photoluminescent CDs from a solution of bovine serum albumin and urea via a microwave treatment process and applied them as pH and temperature nanosensors. The protein carbon source can be responsive to a specific type of metal ions [215]. Medeiros et al. [216] studied the bottom–up methods for producing CQDs, by focusing specifically on microwave irradiation technology, and investigated different applications of their obtained CQDs. This process can produce hydrophilic, hydrophobic, and amphiphilic CQDs [216]. Manioudakis et al. [217] produced hydrophilic CQDs through a microwave-assisted reaction between nitrogen (passivating molecules) and citric acid. They studied the role of N-doping on CQDs’s photo-physical properties [217]. The size of the obtained CQDs was 2.5 nm and the distribution was narrow. The FTIR and XPS results indicated the final product had good dispersibility in water due to the presence of functional groups (carboxylic acid, amine, etc.) on the surface of CQDs [217]. Ang and coworkers [218] used palm kernel shell biomass as a precursor to generate CDs via a microwave irradiation method and studied different parameters such as irradiation period, presence of chitosan, and reaction medium. The quantum yield for obtained carbon dot from diethylene glycol with a 1 min irradiation period was 44%. They concluded the final CDs can be applied in cellular imaging and for removing heavy metal ions [218]. Shejale and coworkers [219] synthesized N-CQDs via microwave pyrolysis technology, and applied them as a co-photoactive layer with an 8.75% photo-conversion efficiency. Also, the power conversion efficiencies were 55% and 99% for co-sensitizer and sensitizer, respectively [219].

Thermal Decomposition: Another standard bottom–up method for the synthesis of CQDs from small organic molecules is thermal decomposition. In this method, smaller carbon units (organics) convert to semi-conductor and magnetic nanomaterials under external heat with sizes less than 10 nm which classifies them as CQDs. This strategy of decomposition reactions is either irreversible such as the decomposition of proteins or reversible such as the decomposition of ammonium chloride [220]. The advantages of this approach are simplicity of operation, lesser time consumption, and inexpensive and large scalable production capability [185,221]. Wang et al. [222] synthesized CDs from citric acid as the carbon source by using thermal decomposition. They passivated citric acid with organosilane and N-(β-aminoethyl)-γ-aminopropyl methyl dimethoxy silane (AEAPMS) during the process. The reaction mixture was heated at 240 °C for 1 min. Finally, they synthesized the CDs with diameters of about 0.9 nm. Wang et al. [223] used different conditions for synthesis of CDs through thermal decomposition. In this approach, they put citric acid on a hot plate at 200 °C for 30 min and then, neutralized it with sodium hydroxide solution, and finally purified it through dialization. This product had a size range from 0.7 to 1 nm. Martindale et al. [224] could synthesize CQDs with an average size of 6.8 ± 2.3 nm and high yields of around 45% through pyrolysis of citric acid at 180 °C for 40 h. Ma et al. [221] could convert Ethylene diamine tetraacetic acid (EDTA) to N-doped GQDs by using a sand bath at 260~280 °C through direct carbonization. These products have graphene-like structures. Table 6 shows some research that has synthesized CQDs through different methods, including the synthesized method, precursor, average size of CQDs, and crystallinity.

#### 2.2.3. Functionalized CQDs

Low stability and biocompatibility of higher doses of CQDs at physiological conditions are some of the main challenges of applying CQDs in biomedical applications. In this regard, functionalizing CQDs can be introduced as a solution method to improve bactericidal, biocompatibility, and biodegradability properties [243]. The surface of CQDs could be functionalized with different surface groups. The presence of some functional groups such as hydroxyl and carboxyl on CQDs’ surface will result in improved water solubility. The functional groups on CQDs’ surface not only increase solubility but also cause the formation of stable colloids in aqueous/polar organic solvents which is advantageous in comparing GQDs that have a weak solubility in ordinary solvents [244].

Zhang et al. [245] functionalized CQDs with PEI and hyaluronate (HA) for tumor targeting and gene delivery. The synthesized CQDs displayed excellent dispersibility and desirable biocompatibility in aqueous solutions and were internalized easily into the cytoplasm of cancer. This nanostructure was applied in intracellular imaging and gene delivery systems due to its extraordinary photoluminescence features. Xiang-Yi [246] synthesized CQDs by functionalizing them with polyethyleneimine (PEI) through a single-step hydrothermal method and using biomass tar as a carbon precursor. The results indicated that the final structure of CQDs consisted of spheres with uniform sizes and the dispersibility in aqueous solution had improved due to the existence of functional groups on the surface of functionalized CQDs. Also, the fluorescence quantum yield of CQDs was enhanced up to 27.3% after being functionalized with PEI [246]. Ghosh and coworkers [247] fabricated CQDs from sweet lemon peel and attached them to various generations of polyamidoamine (PAMAM) dendrimers, then conjugated them to the RGDS peptide to target integrin, which is over-expressed in Triple-negative breast cancer (TNBC). the results of cellular cytotoxicity, DNase I assay, hemolysis assay, cellular uptake, and in vitro transfection confirmed that this nanocarrier is a promising gene carrier system for TNBC gene therapy [247].

CQDs are generally acknowledged for their biocompatibility owing to their carbon-based composition. Nevertheless, their potential toxicity is subject to multifaceted influences such as size, surface functionalization, and concentration. Notably, smaller CQDs may manifest heightened toxicity due to augmented surface area, amplifying interactions with biological systems [248]. The pivotal role of surface modifications in dictating toxicity is underscored, with judicious functionalization augmenting biocompatibility and mitigating adverse effects. Elevated concentrations of CQDs pose the risk of inducing cytotoxicity, necessitating optimization for specific applications to mitigate potential harm. In vivo studies are imperative to unravel the physiological response to CQDs, with a focus on organ-specific toxicity and long-term effects. Strategies encompassing surface engineering, size modulation, and development of biodegradable CQDs, along with meticulous in vitro and in vivo assessments, contribute to refining toxicity profiles [249]. Conjugating CQD surfaces with biocompatible entities, such as polymers or biomolecules, not only enhances stability but also mitigates potential toxicity. The precision in controlling CQD size through adept synthesis techniques allows tailored applications, favoring larger CQDs for certain purposes to assuage toxicity concerns. Pioneering the realm of biodegradable CQDs assures their metabolic breakdown and subsequent elimination from the body, addressing long-term toxicity apprehensions [250]. A holistic evaluation of toxicity, spanning in vitro cell studies and in vivo animal models, is imperative to gauge the safety parameters of CQDs. Progressing from preclinical investigations to clinical trials is pivotal, affording a comprehensive understanding of CQD behavior in human systems and potential toxicity issues [251].

#### 2.2.4. Application of CQDs in Gene Therapy

As mentioned before, CQDs have a high potential for application in cancer imaging, a cancer diagnosis in early stages, virus sensing, and cancer therapy due to their large surface area to volume ratio, photoluminescence, low toxicity, eco-friendliness, good water solubility, effective conjugation to organic/inorganic materials and biocompatibility [252]. Lo and coworkers [253] modified the structure of GQDs with polyethyleneimine to improve GQDs’ biocompatibility and their application as cargo for green fluorescent protein (GFP). Also, the new structure of GQDs could attach to colon cancer cells and epidermal growth factor receptors to increase cell membrane penetration and cell uptake of cargos. Şimşek et al. [254] synthesized CDs with 2.05 nm sizes from Nerium Oleander extract via thermal method. Results showed that the generated CDs can penetrate cell nuclei to react with genes resulting in DNA damage [254]. Ahn et al. [255] fabricated N-doped GQDs for the transfection of different genes such as mRNA, and pDNAs. Electrostatic complexes are composed of N-doped GQDs (with a positive charge) and mRNA or pDNAs (with a negative charge) and they act as a vehicle for efficient transfection of genes into target cells. They [255] concluded these N-doped GQDs with high stability and low toxicity can be introduced as a novel platform for gene delivery in the future. Hasanzadeh et al. [256] synthesized zinc/nitrogen-doped carbon dots via a single-step microwave method. These nanocarriers can transfer large plasmids with high yield and also, deliver mRNA into HEK-293 cells. The results indicated that the newly synthesized CDs with notable photoluminescence features and high transfection efficiency can be a promising candidate for the delivery of both clustered regularly interspaced short palindromic repeats (CRISPR) complexes and mRNA [256]. Different functional groups exist on the surface of CQDs, which make them attach to various biomolecules such as antigens, genes, and antibodies. Zhang et al. [257] synthesized CDs from citric acid and panteaethylenehexamine through a single-step microwave-mediated path. Results showed the synthesized CDs were full of amine groups and had a low toxicity. Therefore, the generated CDs could conjugate to DNA non-covalently, and high transfection efficiency could be observed [257]. Alarfaj and colleagues [258] synthesized CQDs from Citrus lemon pericarp via hydrothermal method and attached it to zinc oxide to determine cytokeratin-19 fragment antigene in human serum [258]. Pramanik et al. [259] synthesized CQDs from mango and prune. In the next step, it was conjugated to anti-HER2 antibodies to detect heterogenicity in breast cancer. They concluded that multicolor nano-scale systems synthesized from natural fruits can be used for determining cancer heterogeneity [259]. Wang et al. [260] synthesized CDs from citric acid and tryptophan via a one-step microwave method. The obtained CDs were conjugated to polyethylenimine and used for siRNA delivery to cell line MGC-803 [260]. Liu and coworkers [261] functionalized GQDs with polyethylenimine and applied it in the miRNA delivery system. The results indicated that the new structure of GQDs was able to deliver virgin and functional mRNA to Huh-7 hepatocarcinoma cells [261]. Huang et al. [262] synthesized photostable CQDs from degraded products via hydrothermal treatment. The size distribution of generated CQDs was in the range of 2–6 nm and the quantum yield was 13%. Huang et al. investigated the optical bio-imgaing capability of synthesized CQDs at in vivo scale [262]. Optical time elapsed imaging results of CQD-wheat straw distribution in a tumor-bearing mouse and the results of optical fluorescence intensities within the harvested organs are shown in Figure 4a,b, respectively [262]. Finally, they concluded the final CQDs could be applied in bio-imaging and phototherapy [262]. Hua and coworkers [263] fabricated a series of nucleolus-targeted red emissive CDs. The results showed Ni-para-phenyl-diamine CDs (Ni-pPCDs) are the best choice between all synthesized CDs, due to high photo-stability, excellent water dispersibility, high quantum yield, and polarity dependent fluorescence emission [263]. Figure 4c,d shows the confocal and STED images of A549 cell, which indicate enlarged nucleoli with a high imaging resolution (146 nm) [263]. In addition, Ni-pPCDs were used for in vivo imaging in both zebra-fish and mice models. They concluded CDs have great potential for application in bio-imaging [263]. Table 7 indicates some studies regarding application of CNTs in gene therapy.

Wu et al. [273] investigated the unwanted effects of intrahippocampal injections of 3-mercaptopropionic acid (MPA)-functionalized CdTe QDs, ultra-structure of neurons, hippocampus, and synapses in rats. Transcriptome sequencing was applied to discover the underlying mechanisms [273]. The results of histological analyses of the hippocampus of rats treated with 2.2 nm and 3.5 nm MPA-capped CdTe QDs showed an irregular arrangement of neurons and cellular swelling [273]. Wang et al. [274] synthesized polymer-coated nitrogen-doped carbon nanodots (pN-CNDs) with particle sizes between 5 to 15 nm through a solvo-thermal reaction. The original product showed high water solubility and adjustable fluorescence. Figure 5b (A–E) shows the fluorescent imaging at different times of post-injection of pN-CNDs [274]. Figure 5b (F,G) shows coronal imaging of the main organs 90 min after pN-CND administration. The results showed glioma possesses a high and considerable accumulation of pN-CNDs [274]. Liu et al. [275] reported that the produced CDs irradiated with light will convert to toxic molecules either for normal (HEK-293) or cancerous (HeLa and HepG2) human cells. Figure 5c–j shows the effect of different parameters such as light intensity, pH, wavelength, temperature, and ionic strength [275]. The results showed CD degradation increased as light intensity increased (Figure 5c) and the wavelength decreased (Figure 5d). Also, more CDs degraded at higher temperature or pH (Figure 5e,f) [275]. Ionic strength (Figure 5g) and size (Figure 5h) did not play an important role in CD degradation. Liu et al. reported [275] that when CDs were irradiated with white fluorescent for 10 min, alkyl and hydroxyl radicals were organized (Figure 5i). Figure 5j indicates the main role of isopropanol in decreasing degradation of CDs [275].

### 2.3. Nanodiamonds (NDs)

#### 2.3.1. Structure and properties of NDs

Diamond nanoparticles were first synthesized via the detonation technique in 1960 [276]. NDs are based on carbon building blocks with a sp^3^-hybridization center covered by sp^2^ carbon atoms [277]. In other words, it is made from a diamond core and amorphous carbon layers. High surface area, sp^3^ carbon matrix, and the presence of functional groups on their surfaces. These features enable nanodiamonds to adsorb biomolecules, act as drug carriers, and modulate cellular signaling pathways. Furthermore, the diamond core can serve as a biocompatible scaffold for various therapeutic molecules, making nanodiamonds a versatile platform for drug delivery and imaging applications [278]. Their diameter is usually less than 20 nm [277]. NDs and synthetic diamonds are relatively cheap. Their unique and inherent properties such as superior hardness and Young’s modulus, optical properties and fluorescence, resistance to harsh environments, high thermal conductivity, and electrical resistivity have attracted remarkable scientific attention for various applications [279]. NDs are also intrinsically biocompatible with low toxicity, stable fluorescence, high thermal conductivity, and large surface areas, which makes them suitable for biomedical applications such as drug/gene delivery [280], bio-imaging, and biosensors [281]. As delivery systems, NDs are capable of transferring a broad range of therapeutic materials such as proteins, small molecule inhibitors, peptides, and chemotherapy drugs. NDs can withstand high dosages in various ranges of cell lines and animal models that confirms their low toxicity. Biodistribution studies have indicated that NDs do not accumulate in vivo for a long period of time [282]. The high surface area of NDs is suitable for the conjugation of molecules. Various purification processes and surface modification affect the biocompatibility of NDs, and extensive studies have also been performed on the toxicity of NDs [283,284]. Understanding the mechanisms by which nanodiamonds interact with biological entities is crucial for harnessing their full potential in various biomedical applications. The nano-bio interface serves as the critical interface where nanodiamonds interface with biological entities, including cells, proteins, and nucleic acids. The interplay at this interface is governed by complex physical and chemical interactions, such as electrostatic forces, van der Waals interactions, and hydrogen bonding. Surface modifications of nanodiamonds play a pivotal role in modulating these interactions, allowing for tailored bio-functionalization and improved specificity in targeting biological components [285]. The dynamics of nanodiamond interactions with cells involve intricate physico-kinetic processes. Cellular uptake mechanisms, intracellular trafficking, and eventual fate within the cellular milieu are influenced by factors such as nanodiamond size, surface charge, and functionalization. Understanding these physico-kinetic aspects is paramount for optimizing nanodiamond-based technologies, including drug delivery systems and diagnostic probes [286]. Schrand et al. studied the cytotoxicity of NDs by detecting the function of mitochondria and the yield of adenosine 5′ triphosphate [287]. They reported that the cell activity was more than 95% and NDs did not show any considerable toxicity in different cell types [287]. Fryer and coworkers [288] evaluated the cytotoxicity and fluorescence imaging capacity of nitrogen-vacancy NDs with sizes in the range of 40 to 90 nm. Results showed that as the size of NDs increased, the cytotoxicity of NDs decreased and NDs with size 90 nm indicated high cellular uptake and fluorescence intensity [288]. Karpeta et al. [289] investigated the cytotoxicity of NDs in different cell lines and studied NDs’ biological activities such as cell viability, lipid peroxidation, and level of reactive oxygen species. Results showed the accumulation of NDs in skin tissue was more than that in other organs in C57 mouse. They concluded that NDs can play a significant role in diagnosis of skin cancer [289]. NDs have a high tendency to agglomerate in aqueous solutions. Sonication and mechanical grinding technology are two solution methods for overcoming this issue. However, surface modification with biocompatible polymer and biological molecules has been introduced as an effective method to change the biological and physicochemical properties of NDs [290]. NDs agglomerate easily and disperse weakly in aqueous solutions. Dispersion of NDs is possible through mechanical and sonication processes. Maitra et al. [291] investigated the effect of different surfactants such as Sodium bis(2-ethylhexyl) sulfosuccinate, Triton X-100, Cetyltrimethylammonium bromide, tert-octylphenoxy poly(oxyethylene) ethanol, and polyvinyl alcohol on dispersion property of NDs. Results indicated that tert-octylphenoxy poly(oxyethylene) ethanol has the best dispersion in water at a minimum concentration of surfactant [291]. The surface of NDs can be functionalized/modified with various ingredients and different functional groups such as a hydroxyl group (-OH), carboxylic acid group (-COOH), sulfur group, amino and ester groups created in NDs’ structures; among them, carboxylic acid groups (-COOH) are the most widespread. Functionalizing NDs will improve their dispersibility, biocompatibility, and make them to conjugate to other materials/biological molecules to form nanostructures with specified physicochemical properties for different selective applications [292].

#### 2.3.2. Synthesis Methods of NDs

NDs can be synthesized by different methods with each of the synthesis techniques affecting their physical and chemical properties. The size, shape, surface structure, and quality of pristine NDs are a function of synthetic methods. NDs’ synthesizing methods are large ablation, high-pressure high temperature (HPHT) [279], plasma-assisted chemical vapor deposition (CVD) [279], autoclave synthesis from supercritical fluids, ion irradiation of graphite, electron irradiation of carbon anions and ultrasound cavitation [293,294]. Two commercial-scale NDs exist in the market today. First, NDs are synthesized through detonation processes with a size range of 4 to 5 nm called detonation NDs, and then NDs are synthesized through the HPHT option with an average size of more than 30 nm. In the following, CVD, HPHT? and detonation techniques are discussed in detail.

Chemical Vapor Deposition Method (CVD): CVD is a procedure for producing inorganic materials, and applies one or more gas-phase combinations including the main product compound to accomplish a chemical reaction on the substratum surface that produces the original products. Applying this technology to generate diamonds started originally in 1970 [295]. By applying this method, large-size diamonds with high purity can be synthesized. NDs synthesized via the CVD method do not require a supplementary purification stage, and the density of the luminescent centers can be selected which is useful for optical research. Besides the positive attributes of the CVD method for synthesizing NDs, there are some disadvantages such as high cost, low deposition rate, being toxic and flammability of reaction sources and residues that limit using this method [292]. The CVD method is one of the most classic methods for the synthesis of ND films [279]. ND films are classified into two categories based on their microstructure and growth environment: the Ultra-nanocrystal diamond (UNCD) and nanocrystalline-diamond (NCD) [296]. Ultra-nanocrystal diamond (UNCD) films are usually grown in a CVD environment that is argon-rich and hydrogen-poor. Their microstructures look like crystalline grains (2–5 nm size) with a layer of ND carbon around them. This microstructure contains up to 2–5% sp^2^-bonded carbon content [296,297]. Hexane/nitrogen-based CVD is applied to produce NCD. Initial nucleation sites for generating this type of film are not needed. This type of film was grown on Ni substrates in a microwave plasma reactor at a temperature range of 400–600 °C [297]. Some forms of NCD can be generated from the decomposition of a gas mixture (usually methane and hydrogen). In this method, radicals such as ${\mathrm{CH}}_{3}^{•}\;$ and $H^{•}$ will be formed, which are intransitive for diamond growth. NCD film can be formed on a silicon wafer coated with diamond powder and acting as a site for NCD nucleation. Finally, a continuous film can be formed as an NCD film [296]. Feudis et al. [298] produced NDs with massive brightness via the CVD method without requiring any seeded substrate. Optical results at low temperatures confirmed this CVD method can generate NDs with proper dispersibility, which can be applied in quantum technologies [298].

High Pressure—High-Temperature method: HPHT is another main method for the synthesis of NDs that was introduced by Wentorf for the first time in 1965 [299]. No metal impurities were observed in NDs produced from organic compounds. These NDs are very biocompatible. High toughness, thermal stability, and, wear-resistance are three considerable properties of NDs synthesized through the HPHT method [292]. In the HPHT process, pressure and temperature should be almost 6 Gpa and 1500 °C, respectively, to change graphite powder into diamonds in the presence of a catalyst. Applying materials such as fullerenes and CNTs instead of graphite for producing diamonds needs much lower temperature and pressure [300]. CNTs can be converted to diamond at 1300 °C and 4.5 Gpa in the presence of NiMnCo as a catalyst. NDs’ size can be controlled accurately when synthesized from chloroadamantane molecules via HPHT technology [301]. Silicon and nitrogen-doped NDs were successfully manufactured by the HPHT method. Davydov et al. [302] suggested a novel technology for synthesizing NDs based on a blend of organo-silicon compounds, hydrocarbons, and, fluorocarbons at high pressure. Small crystal diamonds were formed in the shape of titanium capsules with volumes less than 2 mm^3^ at a pressure of 12–13 Gpa and a temperature range of 1300–2300 °C [303]. Onodera et al. reported another synthesis method with softer conditions (pressure of 6–9 Gpa and temperature of up to 1300 °C) for producing nano-crystalline diamonds in 1992 [304]. In this process, the tantalum capsule is used as a catalyst for transforming graphite into diamond. Ekimov et al. [305] found an optimum condition for large-scale synthesis of nano- and micro-crystal diamonds from adamantane (a type of attractive hydrocarbon for the synthesis of micro- and nano-crystal diamonds) by this method. They synthesized samples in titanium capsules at pressures of 7.7 and 9.4 Gpa and a temperature range of 300–1600 °C. They concluded that a pressure of 9.4 Gpa and temperature of 1250–1330 °C are the best operating conditions for large-scale synthesis. Also, they reported that the synthesization of ND is a direct result of adamantane carbonization [305]. Liu and coworkers [306] synthesized NDs from naphthalene through a two-step HPHT method without the need for a metal catalyst. They observed the grain sizes of NDs with 35 min carbonization time were so heterogeneous, and the grain sizes for NDs with 90 and 135 min carbonization time were smaller and homogeneous [306]. They observed that the average grain size reduced as carbonization time increased [306].

Detonation Technique: The detonation method is clearly more different when compared with other techniques. The explosive impact method is the first method for producing explosive materials. In this process, a phase transition happens outdoors on the graphite precursors and converts to diamond under shock wave pressure and high-temperature conditions. Shock compression of graphite and detonation of explosive materials under lack of oxygen are two technologies for generating NDs. NDs were first generated via detonation technology in the USSR in the 1960s [292]. Greiner and coworkers produced NDs with a size range between 4 to 7 nm through detonation technology [307]. Easy operation, simple instruments, speedy reaction rate, and inexpensive synthesis cost are some of the advantages of this method for producing NDs. Also, NDs synthesized through the detonation process have excellent properties such as good chemical stability, ultra-high hardness, large surface area, high biocompatibility, and low resistance [292]. Different functional groups such as -COOH, -OH, -SO_4_, and, -Cl may exist on the surface of synthesized NDs through the detonation technique, which changes the NDs’ surface potential and causes agglomeration of NDs. The main drawback of this technique is that the resulting NDs include graphite, amorphous carbon, and metal impurities, which are observed in the solid-phase products generated via the detonation method. Consequently, purification and dispersion processes are two major and important steps to remove the impurities from NDs synthesized via the detonation technique [292]. Sonication, ball grinding, or application of these two methods are subcategories of mechanical dispersion methods. Liquid phase reaction and gas phase treatment are two subcategories of chemical dispersion methods [308]. Yoshikawa et al. [309] evaluated non-functionalized/functionalized detonation NDs via dynamic light scattering and cryogenic transmission electron microscopy. They [309] concluded the non-functionalized NDs are dispersed in aqueous suspensions through the long-range and weak electrical double-layer repulsive interaction, while the NDs functionalized with polyglycerol are dispersed in short-range and strong steric repulsive potential barrier produced by polyglycerol. Makino and colleagues [310] evaluated the practical scale fabrication of NDs containing group IV-vacancy centers (silicon-vacancy (SiV), tin-vacancy (SnV), germanium-vacancy (GeV)) through a detonation process. These type of NDs are capable to be applied in bio-imaging and sensing as as fluorescent markers [310].

#### 2.3.3. Functionalized of NDs

NDs in their pure form cannot be directly applied for pharmaceutical applications and therefore, need some type of modification [311]. Surface functionalization is one of the modification methods, which is implemented by the introduction of chemical materials on the surface of NDs to induce it to acquire the favored characteristics [312]. Pure NDs have a burly quantity of impurities and also carry big and diversified functional groups, which create non-uniform surface characteristics. Pure NDs have lower penetration efficiency which restricts intracellular drug delivery. Negative zeta potential, low biocompatibility and stability, lower loading efficiency, and incompetent distribution restrict the applicability of NDs as a therapeutic nanocarrier [313]. To overcome these obstacles and explore NDs as potential nanocarriers, the application of the surface functionalization method on NDs has been proven to be useful to a large extent. Surface functionalized NDs have been revealed to possess appropriate features suitable for applications in diagnostic and therapeutic domains. Initial surface termination and immobilization of functional groups onto homogenized NDs are two stages of surface functionalization of NDs. It is recommended to perform initial surface homogenization before the ultimate surface functionalization to provide a sole functional group on the surface. The subsequent stage of surface homogenization includes the attachment of favorable chemical parts/therapeutic ligands on NDs. This additional attachment can be obtained by both covalent and non-covalent linkages [314,315]. A covalent bond is a chemical bond that is strong with high energy which cannot be easily broken and provides a stable balance of attractive and repulsive forces between atoms. Covalent functionalization of NDs may be an outcome of the straight interaction of functional molecule and the NDs’ surface or in particular occasions, a linker molecule can be applied to create a linkage between a molecule and the ND’s surface. Some tight linkages can avoid undesirable orientation of the compound on ND’s surface and also stop changes in the 3D structure. Covalent functionalization is mostly utilized for the attachment of different functional groups and polymers on the NDs’ surface which can eventually be applied as a platform for employing NDs as new healthcare nanomaterials [316]. From among the functional groups, anhydrides and carbonates are more appropriate for drug delivery applications, where the molecule is required to be released from the NDs’ surface due to their inclination towards hydrolysis. The physicochemical features of NDs can be strengthened after surface functionalization. Functionalizing NDs increases their solubility compared to pure NDs. The improved solubility of functionalized NDs will facilitate biomedical applications by conjugation of more complex molecules such as proteins [317]. Xing et al. [318] investigated the cytotoxicity of various forms of NDs by functionalizing with oxygen atoms covalently. The results specified that oxidized NDs have less cytotoxicity than pure NDs. Hsu et al. [319] functionalized NDs with the carboxylated and hydroxylated groups covalently. In this way, thiol groups on the ND surface were formed and helped to improve cytotoxicity. This complex did not expose the human cell line to any toxicity. Consequently, it provides a novel platform for anti-neoplastic drug delivery. A non-covalent interaction includes electromagnetic interactions between molecules or within a molecule and can be divided into various categories, such as electrostatic, π-π effects, van der Waals forces, and hydrophobic and hydrophilic interactions. Maintenance of structural networks and removing loss of electronic properties are the main benefits of non-covalent linkages [320]. Non-covalent modification of NDs was performed to enhance properties such as hydrophilicity, dual-band electromagnetic absorption, adhesion, and cytotoxicity. The NDs are modified with hydroxyl, carboxyl, and polymeric or organic materials to attain characteristic improvements [321]. Tsai et al. [322] studied the interaction of functionalized NDs with red blood cells. In their research, the surface of NDs was modified with the carboxylic acid groups and conjugated with serum albumin via physical absorption. The results indicated that the surface of NDs was stable for two days. They also indicate that the physical adsorption of NDs does not influence RBC’s oxygenation state. These results suggest that NDs are safe for different biochemical processes without disconcerting the physiological condition of the blood.

#### 2.3.4. Application of NDs in Gene Therapy

NDs are ideal and promising nano-scale carriers of genetic materials due to their adjustable surface chemistry, water dispersibility, high biocompatibility, narrow particle size distribution, semi-spherical shape, and high surface area [323]. Polyethyleneimine (PEI), polyamidoamine (PAMAM), lysine, and polyallylamine hydrochloride (PAH) are the cationic polymers that have been usually used to modify NDs’ surfaces for gene delivery. Those modifications make NDs have a positive charge surface and good distribution. So, the nucleic acids with a negative charge surface could be effectively loaded. However, the safety issue of those functionalizations should be evaluated attentively for clinical usage, because these polymers have not been applied on a clinical scale [324]. NDs can operate as multitasking nanocarriers to transfer genes in biological systems with high delivery yield and high biocompatibility. The output of drug delivery can be grown to 70 times of normal gene delivery. However, more studies are needed to understand the interaction with cells, the cytotoxicity, and the bio-distribution of NDs [325,326]. The requirement of conjugating to other vectors to increase the stability of carbon-based nanostructure is one of the limiting factors in the application of these structures. Niosomes can be used to overcome this issue. They are lipidic vectors, including a cationic lipid, and a non-ionic surfactant, with high gene packing capacity that are capable of compressing, protecting, and releasing genetic materials in a safe way. Also, a helper component can be added to the system to improve the biological activity potential of the system. Therefore, nanodiasomes which are formed from a combination of NDs and niosome are non-viral vectors for use in gene delivery systems [323]. NDs were functionalized with low molecular weight PEI and applied for plasmid and siRNA delivery. Results showed that ND-PEI complexes have a high capability for in vivo gene delivery. ND-PEI complexes are able to perform endosomal escape if their surface is coated completely with many amino groups, and this favors intracellular siRNA delivery [327,328]. Xu et al. [329] fabricated NDs-siRNA complexes through electrostatic interaction. Results showed that siRNA delivery by NDs into tumor cells is more effective and higher than liposomal formulations and, is able to decrease proliferation of tumor cells [329]. In addition to siRNA, Liu et al. elaborated nanodiamond-based microRNA delivery system that promotes pluripotent stem cells toward myocardiogenic reprogramming [330]. Nanodiamond-based microRNA delivery system promotes pluripotent stem cells toward myocardiogenic reprogramming [330]. Al Qtaish and coworkers [323] prepared nanodiasomes and studied their capability as a non-viral carrier in gene delivery systems. Results showed the transfection efficiency in HEK-293 cells and biocompatibility value increased, and biological results confirmed the complex of NDs and niosomes had been applied for the treatment of central nervous system diseases [323]. UNC0646 is a small molecule EHMT2 inhibitor (euchromatic histone lysine N-methyltransferase 2) with a good efficacy at in vitro scale, but its efficiency is not satisfactory at in vivo scale, due to poor water solubility. This inhibitor is used for hepatocellular carcinoma treatment [331]. Gu et al. [331] used NDs as a vector for inhibitor delivery at in vivo scale. They synthesized ND-UNC0646 complexes with desired drug delivery features through physical adsorption, and observed that the dispersibility of UNC0646 inhibitor improved at in vivo scale, and it can be applied for intravenous administration [331]. Figure 6a shows the tumor growth results by IVIS imaging system. They observed that the ND-UNC0646 complex with a high dosage (2 mg/kg) had a considerable tumor inhibiting efficiency. Therefore, the remedial efficacy of ND-unc0646 complex was a function of dosage [331]. Figure 6b,c taken by Gu et al. indicate the orthotopic liver transplantation surgery and the developed orthotopic liver tumor in mouse [331]. While NDs hold immense potential for biomedical applications, several challenges remain, including a comprehensive understanding of long-term biocompatibility, potential toxicity, and large-scale synthesis methods. Future research should focus on elucidating these aspects to facilitate the translation of nanodiamond-based technologies from the laboratory to clinical settings. This review provides a comprehensive overview of the current understanding in this field and highlights avenues for future research, laying the groundwork for the continued development and application of nanodiamonds in cutting-edge biomedical technologies [332].

## 3. A Perspective on other CBNs

Researchers Since Novoselov and coworkers [333] received the Nobel Prize in Physics for the discovery of graphene in 2004, research on the potential applications of this faltering material has increased exponentially, ranging from electronic, optoelectronics, biomedical engineering, tissue engineering medical implants, and sensors. Beyond the aforementioned applications, the use of graphene for biomedical applications is a relatively new area with significant potential. The first biomedical study on graphene was reported by Sun et al. in 2008 [334] as an efficient nanocarrier for drug delivery. Since then, a lot of interesting works have been carried out to explore the use of graphene in medicine. The intensive research on the bio applications of graphene and its derivatives is due to many fascinating properties of this material, such as high specific surface area (2630 m^2^/g), exceptional electronic conductivity (mobility of charge carriers, 200,000 cm^2^V^1^s^1^), thermal conductivity (~5000 W/m/K), mechanical strength (Young’s modulus, ~1100 Gpa), intrinsic biocompatibility, low cost and scalable production. The chemical structures of graphene, graphene oxide (GO), and redudef form of GO show in Figure 7.

In particular, GO, a precursor for graphene, is endowed with unique features, such as sp^2^ π-π interaction, easy synthesis, high water dispersibility, good biocompatibility, easily tunable surface functionalization, and convenient, inexpensive, and scalable production. GO is more widely used than graphene for biomedical applications due to the presence of carboxylic, epoxy, and hydroxide groups, which permit a wide range of functionalization opportunities. These properties allow GO to be a principal part of drug delivery systems by providing a wide range of functionalization options with a number of water-soluble and biocompatible polymers such as polyethylene glycol (PEG) [335,336,337,338,339,340,341,342,343,344,345,346,347,348]. GO has been used as an efficient nontoxic nano vehicle for siRNA delivery. Zhang et al. [349] delivered siRNA using polyethylenimine (PEI)-grafted GO to enhance chemotherapy efficacy. As a preliminary study, the researchers used the Bcl-2-targeted siRNA as a model. PEI was covalently linked to GO by using a 1-ethyl-3-(3-dimethylaminopropyl) carbodiimide (EDC) chemical reaction. According to their results, the PEI-GO hybrid gene carrier permitted effective loading of siRNA, and its biocompatibility was highly improved compared to PEI alone. In short, they demonstrated that PEI-GO was an excellent nanocarrier for effective delivery of siRNA. All these attributes also make GO an interesting candidate material, also for gene delivery purposes. Strong π-π stacking interaction between GO and single-stranded DNA (_SS_DNA) facilitates the loading of _SS_DNA onto GO and therefore protects _SS_DNA against enzymatic digestion. However, no such interactions occur between GO and double-stranded DNA (_ds_DNA), as DNA bases are involved in base-pairing within the double helix, preventing the generation of π-π stacking interactions between GO and _ds_DNA. In addition to the poor loading efficiency, it was found that GO is not stable in simulated physiological solutions, such as phosphate-buffered saline (PBS). Therefore to enhance the loading efficiency of plasmid DNA onto GO and to improve the colloidal stability of GO in simulated physiological solutions, chemically modified GO conjugates are necessary.

Different strategies have been adopted by the research community to link cationic polymers to GO and improve its properties as a gene delivery carrier. Basically, there are two strategies to synthesize GO-cationic polymer conjugates. One strategy takes advantage of the negative charges of the carboxyl groups in GO and their ability to interact electrostatically with cationic primary amine groups of the polymer, known as layer-by-layer (LBL) assembly process. For example, upon electrostatic interactions, much safer and more stable GO-PEI conjugates have been formed by Feng et al. [350]. The other strategy to obtain modified GO conjugates involves covalent coupling GO’s carboxyl groups with the primary amine groups of the polymer by EDC/NHS chemistry. In this respect, the resulting vectors have been successfully employed in vitro to deliver plasmid DNA. Among the most employed is the combination of GO with cationic branched polyethylenimine (BPEI). This strategy is particularly interesting since BPEI is widely considered as the gold standard polymer for gene delivery purposes, due to the intrinsic properties that enhance cellular uptake and endosomal escape. However, cytotoxicity effects associated with BPEI severely jeopardize its clinical application for gene delivery purposes. In this sense, it has been reported that the combination of GO with BPEI avoids this scenario since GO-BPEI vectors are more efficient to transfect cells and less toxic than the counterpart vectors that are based only on BPEI [336,351]. Additionally, ternary vectors based on PEG-BPEI-GO, complexed plasmid DNA efficiently for photothermally controlled drug delivery. Upon exposure by NIR irradiation, GO enhanced the endosomal shape of polyplexes, leading to high transfection efficiencies without any cytotoxicity effects [350]. These promising results offer reasonable hope to explore novel future applications of GO for gene delivery purposes further, especially for in vivo applications, since most of the gene delivery studies based on GO have been conducted in in vitro conditions.

Three scientists generated a fullerene (FU), called C60, on a small scale by evaporating the graphite by the arc or the laser beam methods or through an electric current in low-pressure argon or helium in 1985. In fact, FU is one of the allotropes of the carbon family of nanostructures. Fullerenes’ structure consists of sp^2^ carbons that exhibit superior chemical and physical features and a highly symmetrical structure with various sizes (C60, C76, etc.) [352,353]. C60 is the most applied fullerene in synthesized composition. It is made of 60 carbon atoms with C5–C5 single bonds, and C5=C6 double bonds. C60 and C70 are generated at 1000 °C and the pulse duration and the concentration have a direct relation. in other words, as the pulse duration increases the concentration enhances [354].

FUs are empty, inert, soluble in organic solvents, and perpetually modifiable. FUs get swiftly distributed to diverse tissues of the body when injected. FUs are widely applied in biomedical applications such as photodynamic therapy, gene, and drug delivery, etc. [355]. Lack of solubility in water and poor solubility in many organic solvents are some barriers to application in the biological field. Consequently, in biological systems, the hydrophilicity of materials is a more important criterion than hydrophobicity, and there are some procedures to improve the hydrophilicity and water solubility. Some of these methods are the ministration of two-phase colloidal solutions, chemical functionalization by attaching hydrophilic materials such as amino acids, carboxylic acids (-COOH), polyhydroxy groups, an amphiphilic polymer, synthesizing fullerene derivatives, fullerene polymers, encapsulation in particular carriers (cyclodextrins, calixarenes, micelles, liposomes, etc.) [356]. In some research, FUs principally cationic ones were applied to deliver small molecules due to their nonimmunological reactions, affordable and high efficacy. In some cases, C60 is capable of generating oxygen species upon exposure to visible light, making it an appropriate candidate for photodynamic therapy [357,358]. The antioxidant actuality of FUs, ease of topical delivery, and their great interactions with epidermal keratinocytes correspond to tremendous abilities to be applied in transdermal delivery and cosmetic applications [359]. Materials with these outstanding features have been frequently applied in diverse forms, such as moisturizers, anti-inflammatory materials, cytoprotective, and anti-melanogenesis agents [360,361]. Moreover, there are some reports on the use of non-covalently adsorbed drugs and FUs (indicating rational stability at ambient temperature); hence, they could offer acceptable and efficient transdermal drug delivery at an optimum concentration. FU with its fantastic structural features is one of the promising candidates in drug delivery. FU is capable of carrying a multiple-drug payload and could decrease many of the side effects of chemotherapy through targeted drug delivery [362,363]. For example, the side effects of doxorubicin related to cardiomyopathy made this compound a candidate for attaching to FU. This complex was investigated at various pH values and the results indicated 100% drug release at pH 5.25. The results confirmed the application of this conjugation for targeted drug delivery and the reduction of side effects. The issue in this approach was that doxorubicin was water-soluble whereas FU was hydrophobic, so ethylene glycol spacers were applied as linkers for conjugating methano-C60 with doxorubicin, increasing the water solubility of this conjugated compound [364]. In another research, spherical nanostructures that were made of amphiphilic fullerenes with hydrophobic zones, called Buckysomes, were used to cover the hydrophilic surface of paclitaxel. This study specified that the proposed complex could successfully increase drug absorption [365]. A novel drug delivery system relying on an ‘on–off’ drug delivery strategy expanded by the attachment of doxorubicin and FU after the conjugation of a hydrophilic shell to the outer surface [366]. This new system is extremely stable in physiological solutions even with a pH of 5.5 in the ‘off’ state; in mutuality, and in the ‘on’ state. The generation of reactive oxygen species (ROS) by FU resulted in two remedy types. First, oxygen species were produced followed by cell death. Second, the fracturing of the ROS-sensitive linkers resulted in the exploding release of doxorubicin [366]. Korzuchetal et al. [367] synthesized two types of water-soluble FU nanomaterial as non-viral siRNA transfection nanosystems called HexakisaminoC60 and monoglucosamineC60. The HexakisaminoC60 fullerene was an impressive siRNA transfection agent and reduced the GFP fluorescence signal remarkably in the DU145 cells. In contrast, the glycol fullerene JK39 was inactive in transfection tests, probably due to its high zeta potential and the formation of a great stable complex with siRNA. Maeda-Maniya and coworkers [16] investigated the successful gene delivery in vitro with tetra (piperazino) fullerene epoxide (TPFE) and its preference for Lipofectin. Further on, they investigated the efficacy of in vivo gene delivery by TPFE. They [16] concluded that the delivery of enhanced green fluorescent protein gene (EGFP) by TPFE on pregnant female ICR mice indicated obvious organ selectivity compared with Lipofectin. Additionally, higher gene expression by TPFE was observed in the liver and spleen rather than the lung. The results indicated that Lipofectin considerably enhanced liver enzymes and blood urea nitrogen. In the end, this research was confirmed to be an effective gene delivery in vivo using a water-soluble FU. It should be mentioned that, in the advancement of new strategies for cancer treatment, particularly in chemotherapy and photodynamic therapy, most of the nanomaterials have been studied so far, but FU and FU-based systems are the most promising candidates owing to their outstanding and distinctive structures and features.

## 4. Future Perspective on Carbon Nanostructure in Gene Therapy

In recent years, the integration of carbon nanostructures into gene therapy has emerged as a promising avenue for advancing the field toward more efficient and targeted interventions. Carbon nanomaterials, such as carbon nanotubes and graphene, possess unique physicochemical properties that make them ideal candidates for therapeutic applications. This review explores the current state of research on carbon nanostructures in gene therapy, highlighting their potential for enhanced drug delivery, gene transfection, and therapeutic efficacy. Furthermore, we delve into the mechanistic insights behind the interaction of carbon nanostructures with biological systems, shedding light on the intricate cellular and molecular processes that underlie their therapeutic effects. As we navigate the intricate landscape of carbon nanostructure-based gene therapy, this comprehensive review aims to provide a forward-looking perspective, discussing anticipated challenges, future directions, and the transformative impact that these innovative approaches may have on the landscape of molecular medicine. By unraveling the intricacies of carbon nanostructure-mediated gene therapy, this review contributes to the ongoing discourse in the scientific community and lays the groundwork for the development of next-generation therapeutic strategies with the potential to revolutionize clinical practice.

Researchers all over the world believe that developments and the profiling of differentnon-viral nanostructures for gene delivery would have a great impact on future gene therapies. In the future, we will observe some of the required high-cost treatments for targeted diseases replaced with inexpensive therapies. New genetic treatments will be created for both ordinary as well as uncommon diseases. Researchers have reported that similar to other biologics, gene therapies are also expected to see notable signs of progress in the coming years.

Nanostructure-based gene delivery systems such as CNTs, QDs, or NDs, are emerging as an exciting new method with the possibility to overcome known obstacles and multiple biological and medical requirements. Despite the fact that they share a common Carbon-based nanostructure, there are relevant differences between them in terms of physicochemical properties, synthesis method, functionalization strategies, and applications in gene therapy that merit the attention of the research community.

The main characteristic of all delivery systems is the need for the transfer of genetic materials into the cells, where their favorable operations will be exposed. Therefore, a requirement for a delivery system is not to let the genetic materials enter the cells passively. The emerging field of nano-gene delivery introduces the utilization of novel nanostructures and their features to generate delivery vectors that can effectively deliver different genetic material into a variety of various types of cells. The physicochemical properties of the carbon-based delivery vectors can be applied to address the ongoing challenges existing in delivering genetic materials at in vitro and in vivo scales. While there is an increasing interest in nanostructure-based gene delivery systems, the string is still in the initial steps, and there is a powerful need to figure out nanostructures and their physicochemical features on biological grounds.

## Figures and Tables

**Figure 1 pharmaceutics-16-00288-f001:**
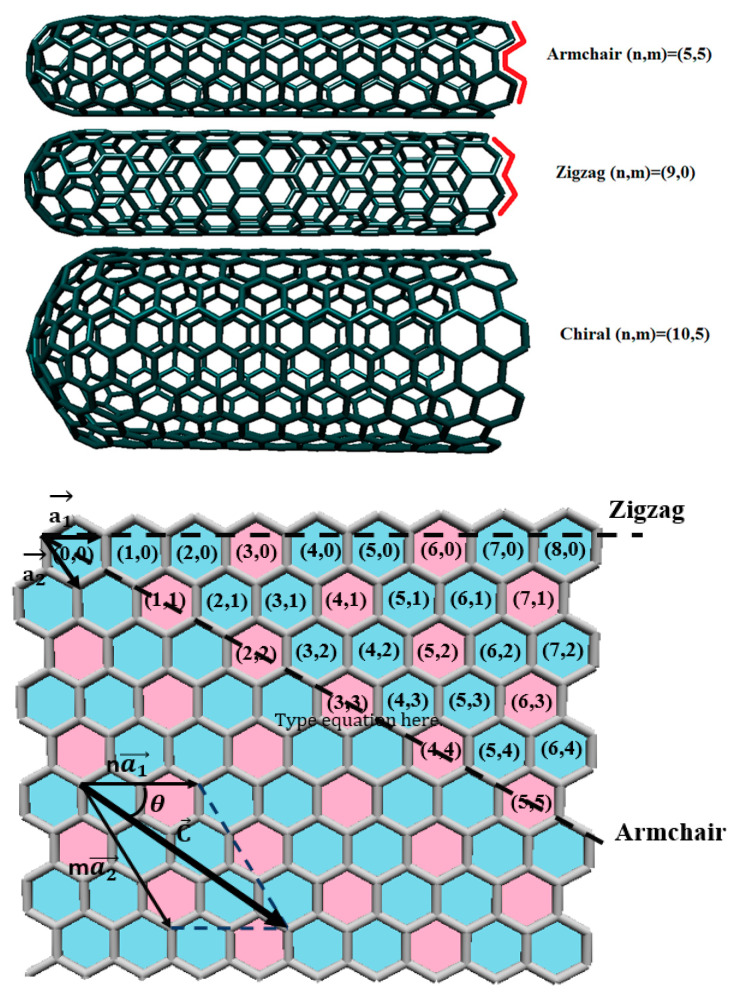
Schematic diagram showing zigzag, armchair, and chiral SWCNTs; the semi-conductive and metallic SWCNTs are labeled with blue and pink colors, respectively.

**Figure 2 pharmaceutics-16-00288-f002:**
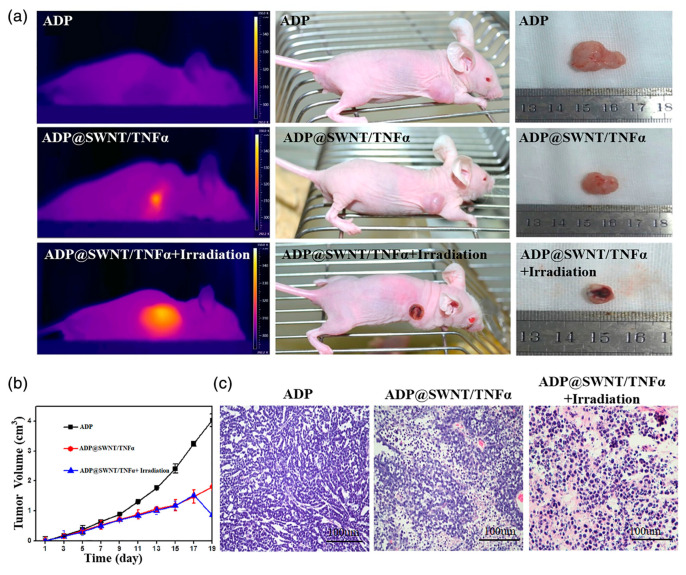
Tumor-inhibitory effects of ADP@SWNT/TNFα and ADP@SWNT/TNFα+laser irradiation on human colorectal cancer in nude mice. (**a**) Photographs, (**b**) tumor volumes (measured every 2 days), (**c**) pathological examination of the gastrointestinal cancer tissues of nude mice injected with ADP, ADP@SWNT/TNFα, and ADP@SWNT/TNFα+irradiation. Reproduced with permission from ref. [134]. Copyright 2022, John Wiley and Sons.

**Figure 3 pharmaceutics-16-00288-f003:**
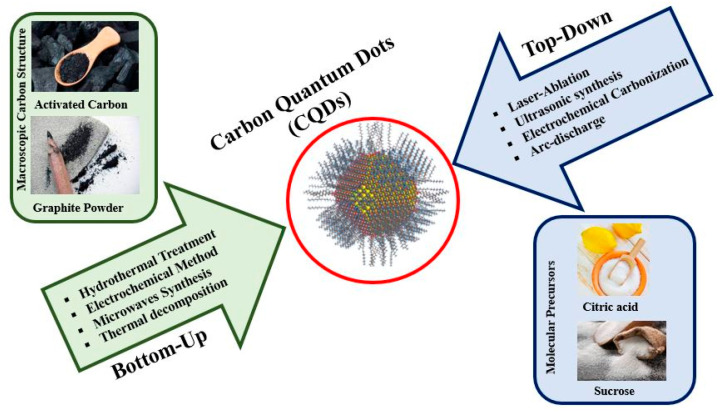
Two main procedures for the synthesis of CQDs.

**Figure 4 pharmaceutics-16-00288-f004:**
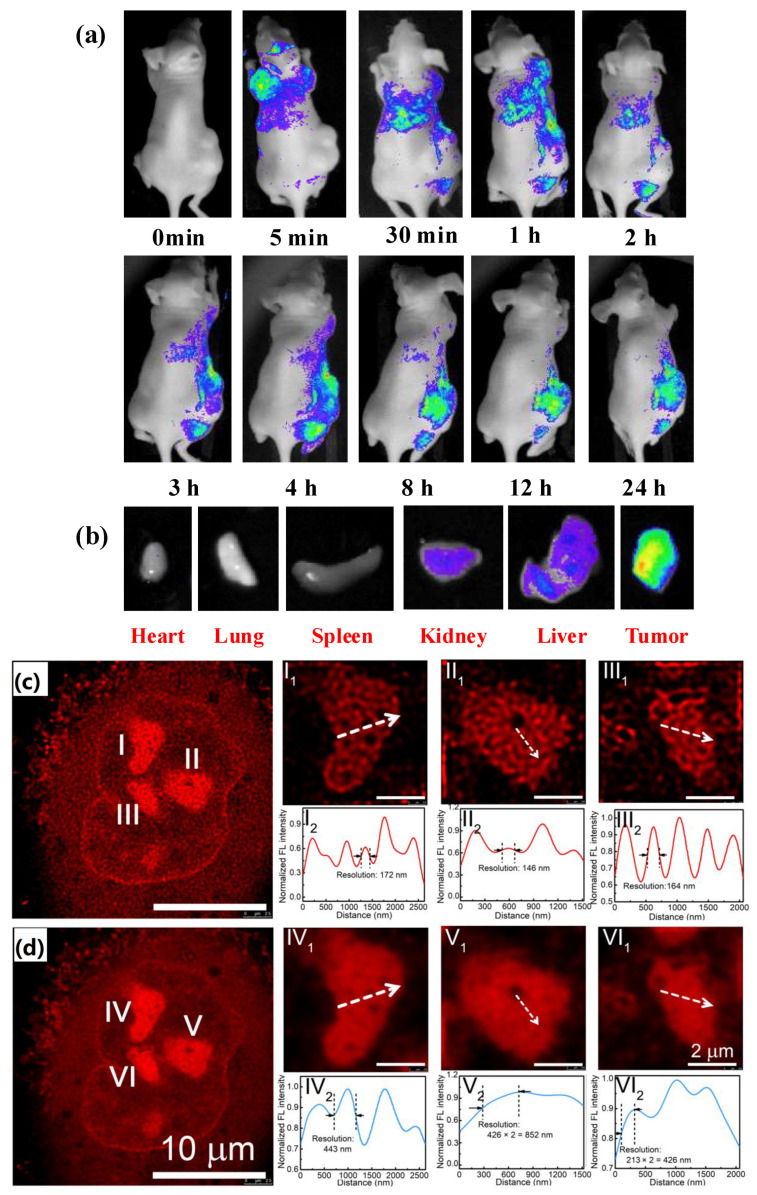
(**a**) In vivo fluorescence imaging of nude mice after intravenous injection of the carbon quantum dot-wheat straw solution; (**b**) representative fluorescence images of dissected organs of a mouse after intravenous injection of carbon quantum dot-wheat straw solution for 24 h. Reprinted with permission from Huang et al. [262]. (**c**) STED image and (**d**) confocal image of a representative A549 cell stained by (Ni-pPCDs). (I1, II1, and III1) Enlarged STED images of the nucleoli of the A549 cell in (**c**), and (I2, II 2, and III2) corresponding fluorescence intensity analysis results of the marked lines in I1, II 1, and III1. (IV1, V1, and VI1) Enlarged confocal images of the nucleoli of the A549 cell in (**d**) and (IV2, V2, and VI2) corresponding fluorescence intensity analysis results of the marked lines in IV1, V1, and VI1. (**c**,**d**) Reproduced with permission from ref. [263]. Copyright 2019, American Chemical Society.

**Figure 5 pharmaceutics-16-00288-f005:**
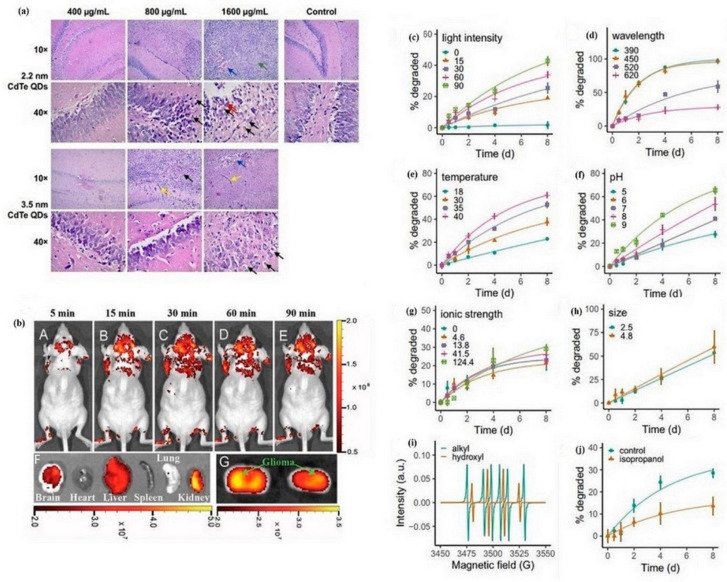
(**a**) Histological analysis of hippocampus treatment of rats treated with 2.2 nm and 3.5 nm MPa-capped CdTe QDs; reproduced with permission from ”*International Journal of Nanomedicine* **2016**, *11*, 2737–2755” Originally published by [273] and used with permission from Dove Medical Press Ltd. Foamy cells are denoted by black arrows, neutrophils are highlighted by red arrows, accumulated quantum dots (QDs) are indicated by blue arrows, necrotic cells are represented by green arrows, and swollen blood vessels are marked by yellow arrows. (**b**) In− and ex−vivo imaging of glioma−bearing mice intravenously administered with the polymer-coated nitrogen−doped carbon nanodots (pN−CNDs). ((**b**), A–E) Whole-body imaging of the pN−CNDs at various time points post−injection. ((**b**), F) Ex-vivo imaging of major organs 90 min after pN−CNDs administration. ((**b**), G) Coronal imaging of the brain 90 min after pN-CNDs administration. The black arrow represents the signal intensity (radiant efficiency) from weak (red) to strong (yellow). Reproduced with permission from ref. [274]. Copyright 2015, John Wiley and Sons. Photodegradation of laboratory-synthesized CDs. (**c**–**f**) More CDs degraded at higher light intensity (0, 15, 30, 60, and 90 μmol photons/m^2^/s) (**c**), shorter light wavelength (390, 450, 520, and 620 nm) (**d**), higher temperature (18, 30, 35, and 40 °C) (**e**), and pH (5, 6, 7, 8, and 9) (**f**). (**g**) Ionic strength (0, 4.6, 13.8, 41.5, and 124.4 mM) had no effects on CD degradation. (**h**) CDs with different sizes (2.5 and 4.8 nm) show similar degradation kinetics. (**i**) Free hydroxyl and alkyl radicals were produced when CDs were irradiated by white fluorescent light (60 μmol photons/m^2^/s). (**j**) When the hydroxyl radicals were scavenged by 10 mM isopropanol, fewer CDs degraded compared to the control treatment without any addition of isopropanol. Data are presented as the mean ± s.d. (n = 3 independent experiments). Source data are provided as a source data file. Reproduced with permission from ref. [275] Copyright 2021, Springer Nature.

**Figure 6 pharmaceutics-16-00288-f006:**
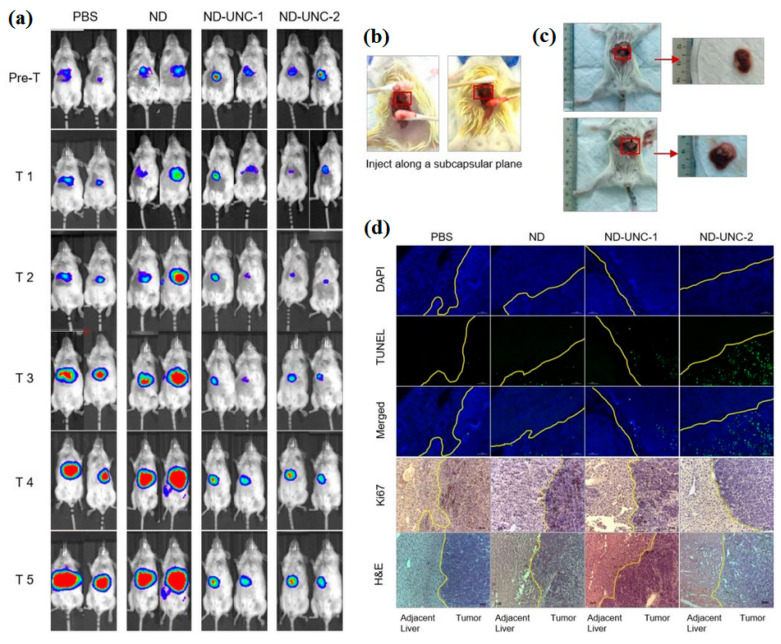
In vivo therapeutic efficacy of ND-UNC0646 evaluated using orthotopic HCC mouse model. (**a**) Representative images of Luc-SNU-398 liver tumor-bearing mice undergoing multiple treatments with PBS, ND, low-dosage of ND-UNC0646 (1 mg/kg), and highdosage of ND-UNC0646 (2 mg/kg) showing in vivo therapeutic efficacy of ND-UNC0646. NDUNC-1 represents a low dosage of ND-UNC0646 (1 mg/kg), while ND-UNC-2 represents a high dosage of ND-UNC0646 (2 mg/kg). ‘Pre-T’ denotes pre-treatment; T 1, T 2, T 3, T 4, and T 5 denote each dose. (**b**) Representative pictures depicting orthotopic liver transplantation surgery. (**c**) Representative pictures showing a mouse that successfully developed an orthotopic liver tumor. (**d**) In vivo biocompatibility assessment by histological analysis. Representative images show apoptotic response and histological morphology in normal liver tissue and HCC tumor tissue after drug treatment. Yellow lines denote the boundaries between adjacent normal liver tissue and tumor tissue. Figure 6 Reprinted with permission from ref. [331]. Copyright 2019, American Chemical Society.

**Figure 7 pharmaceutics-16-00288-f007:**
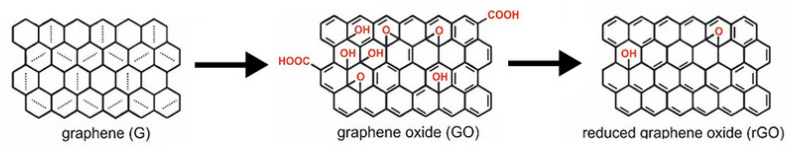
Schematic representation related to the chemical structure of graphene (G), and its precursors graphene oxide (GO) and reduced form of graphene oxide (Rgo).

**Table 1 pharmaceutics-16-00288-t001:** Review of CNTs preparation via the laser ablation method.

CNT Type	Inert Gas	Laser Source	Catalyst	Ref.
SWCNT	Ar	CO_2_ laser	Co, Co/Ni, Fe, Fe/Ni, Ni, and Ni/Y	[64]
MWCNT	Ar	Nd:YAG laser	No metal	[65]
SWCNT	Ar	CO_2_ laser	Co/Ni	[66]
SWCNT	Ar	Pulsed Nd:YAG	Co/Ni	[67]
SWCNT	Ar	Nd:YAG laser	Ni and Co	[60]
SWCNT	Ar	CO_2_ laser	Ni and Co	[68]
SWCNT	N_2_ and B	CO_2_ laser	Carbon	[69]
MWCNT	Ar	Nd:YAG laser	-	[70]

**Table 2 pharmaceutics-16-00288-t002:** Review of CNT preparation via the CVD method.

CNT Type	Inert Gas	Carbon Source	Technique Setup		Catalyst	Ref.
			Time (min)	Temperature (°C)		
MWCNT	Ar	Benzene	5–240	750	Ferrocene	[86]
MWCNT	Ar	Acetylene	60	850	Ferrocene	[87]
MWCNT	H_2_	Methane	10–60	1000	MgMoO_4_	[88]
MWCNT	H_2_	Cyclohexane	5	750	Co, Fe, and Alumina	[89]
MWCNT	N_2_	Polypropylene	40	500–800	NiO/hzsm-5 zeolite	[90]
MWCNT	N_2_	Acetylene	15	750–900	NH_3_/Co	[91]
MWCNT	N_2_	Xylene & Cyclohexanol	-	750	Ferrocene	[92]
SWCNT	N_2_	Ethylene	10	550–750	Fe, Fe/Al, Fe/Al_2_O_3_, and Co/Al_2_O_3_	[93]
SWCNT	Ar/N_2_	Methane	-	1200	Ferrocene	[94]
MWCNT	Ar/N_2_	Ethanol	-	350–750	Cobalt nanoparticles	[95]
SWCNT	H_2_	Eethane	-	550–950	Fe and Alumina	[96]

**Table 3 pharmaceutics-16-00288-t003:** List of covalent-functionalized carbon nanotubes in some research.

Type of CNT	Type of Functionalization	Functionalization Compound	Ref.
SWCNT	covalent	polytyrosine	[105]
SWCNT	covalent	Photosensitizer verteporfin	[106]
MWCNT	covalent	Spiropyran-4’,6’-dicarbonylazides	[107]
Carboxylic CNT	covalent	Hydroxyl-terminated polydimethylsiloxane	[108]
MWCNT	covalent	Imidazolium-based poly (ionic liquid)s	[102]
MWCNT	covalent	Zinc and copper complexes of meso-tetra (4-aminophenyl) porphyrin	[109]
SWCNT	covalent	Methacrylate-co-porphyrins	[110]
SWCNT	covalent	Carboxyl, hydroxyl, and amine chemical groups	[111]
MWCNT	Covalent/non-covalent	Alkaline solution of KMnO_4_	[112]
SWCNT	Covalent/non-covalent	Fluorinated phosphonate analogs of phenylglycine	[113]
SWCNT	covalent	Acyl chloride	[114]

**Table 4 pharmaceutics-16-00288-t004:** List of non-covalent functionalized carbon nanotubes in some research.

Type of CNT	Type of Functionalization	Functionalization Compound	Ref.
SWCNT	non-covalent	Surfactant peptides	[120]
MWCNT	non-covalent	Lectin concanavalin A	[121]
MWCNT	non-covalent	Different deep eutectic solvent compounds	[122]
SWCNT	non-covalent	Pyrene pendant polyester	[123]
MWCNT	non-covalent	Prefunctional polyesters (pyrene, pyrene together with -COOH, -OH, and -C≡CH)	[124]
SWCNT/MWCNT	non-covalent	pyrene-polyethylene glycol derivatives	[125]
SWCNT	non-covalent	Fmoc-amino-acid-bearing polyethylene glycol chains	[126]

**Table 5 pharmaceutics-16-00288-t005:** Overview of the functionalized CNTs to deliver nucleic acid.

Category	Carbon Nanotube Functionalization	Nucleic Acid	Cells	Ref.
Plasmid DNA 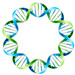	SWCNT-NH_3_^+^MWCNT-NH_3_^+^	pCMV-βgal	A549 cells	[135]
Plasmid DNA	MWCNT-g-PEI	pCMV-Luc gene report	COS7, HepG2, 293 cells	[136]
Plasmid DNA	MWCNT-NH_2_	pEGFPN1	HUVECs, A375	[137]
Plasmid DNA	MWCNT-g-PEIMWCNT-g-PEI/PAA	pCMV-β-gal	A549 cells	[138]
Plasmid DNA	SWCNT-NH_2_	P53 tagged with GFP	MCF-7	[139]
Plasmid DNA	SWCNT-ammonium PEIMWCNT-ammonium PEIMWCNT-CS- FA	pβ-galClontech	CHO cells, HeLa cells, MCF7	[140]
siRNA 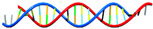	SWCNT-PL-PEG-SS-RNA	Lamin A/C	TC-1 cells, 1 H8 cells, LLC cells	[141]
siRNA	SWCNT-PL-PEG- NH_2_	CXCR4 and CD4 receptors	T-cell line, MAGI cell line, Human peripheral blood, mononuclear cells (PBMCs)	[142]
siRNA	SWCNT-NH_2_	Cydin A_2_	K562 cells	[143]
siRNA	MWCNT-PEI and Pyridinium	Anti-luciferase	H1299 cell line	[144]
siRNA	MWCNT-NH_3_^+^	Caspase-3	N2a cells, Primary cortical neurons	[145]
siRNA	SWCNT-DSPE-PEG-NH_2_	MDM2	B-Cap-37 cells	[146]
siRNA	SWCNT-PEI	hTERT	PC-3 cells	[147]
siRNA	SWCNT-PEI-piperazineSWCNT-aptamer	siRNA	MCF7MDA-MB-231 cells	[131]
siRNA	SWCNT-PEI-CandesartanMWCNT-ammonium cation	siRNA	SiRNA(siVEGF), siRNA (siPLK1)	[148]
siRNA	SWCNT-DSPE-PEG-PEI	siRNA	B16-F10 cells	[130]
siRNA	SWCNT-ammonium PEG	siRNA	HeLa cell, HepG2	[149]
siRNA	SWCNT functionalized with sense and anti-sense strands of a-siRNA and b-siRNA	a-siRNAb-siRNA	mGFP5Nb leaf cells	[150]
miRNA 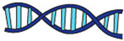	MWCNT-FITC-PAMAM	Antisense miRNA		[151]
miRNA	MWCNT-PEI-g-GNR	miRNA	Hela cells	[152]
miRNA	SWCNT-ammonium PEI/PAMAMMWCNT- ammonium PEI/PAMAM	miRNA	endothelial cells	[132]
AONs 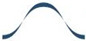	MWCNT-COOHMWCNT-PEIMWCNT-PDDAMWCNT-PAMAMMWCNT-Chitosan	AONs	Hela cells	[153]
AONs	CNT-PLGA	AONs	Osteosarcoma cells	[154]
AONs	SWCNT-COOH	NF-kB	Hela cells, Mononclear cells (MCs)	[155]
AONs	MWCNT-PEG-COOH	AONs	SCLC, DMS53, NSCLC, NCIH2135	[156]
Aptamers 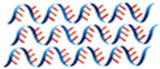	MWCNT-COOH	Aptamers (MUC-1)	MCF7	[157]
Aptamers	MWCNT-PEG-COOHPSMA-aptamer	Aptamer	PC-3 cells	[133]

**Table 6 pharmaceutics-16-00288-t006:** Summary of top–down and bottom–up CQDs’ synthesis methods.

Top–Down or Bottom–Up	Synthesized Method	Precursor(s)	Average Size of CQDs (nm)	Crystallinity	Ref.
Top–down	Laser ablation	Graphite flakes	3.2 nm, 8.1 nm, 13.4 nm	Yes	[225]
Top–down	Chemical Oxidation	g-Butyrolactone	9 ± 6 nm	Yes	[226]
Top–down	Chemical Oxidation	Petroleum coke	CQD: 5.0 nm N-CQD: 2.7 nm	No	[227]
Bottom–up	Hydrothermal	Citric acid and ethylenediamine or N-ethylethane-1,1-diamine or N-(2-aminoethyl) acetamide	In the range of 2–6 nm	Yes/No	[228]
Bottom–up	Hydrothermal	Citric acid, ethylenediamine and ammonia water	~3.7 ± 0.7 nm	Yes	[229]
Bottom–up	Microwave irradiation	Citric acid, urea and formic acid	Green: 0.5–2.0 nm Red: 0.7–2.5 nm	-	[230]
	Chemical vapor deposition	Acetylene	In the range of 2–7 nm	Yes	[231]
Bottom–up	Hydrothermal	o-Phenylenediamine or 4,5-difluoro-1,2-benzenediamine	5.52 nm and 5.18 nm	No	[232]
Bottom–up	Microwave irradiation	o-Phenylene-diamine and citric acid	1.1 ± 0.3 nm	-	[233]
Bottom–up	Pyrolysis	Fennel seeds	3.90 ± 0.91 nm	Yes	[234]
Bottom–up	Hydrothermal	Hydrothermal carbon biomass	100 °C: 2.3 nm, 120 °C: 1.3 nm, 140 °C: 1.9 nm, 160 °C: 2.9 nm, 180 °C: 2.4 nm	No	[235]
Top–down	Electrochemical Carbonization	Graphite rod	6 nm	-	[236]
Bottom–up	Microwave irradiation	Ammonium citrate	~4 nm	-	[237]
Bottom–up	Hydrothermal	Chitosan powders	In the range of 1.5–3 nm	No	[238]
Bottom–up	Hydrothermal	Citrus Lemon Juice	In the range of 1–6 nm	No	[239]
Bottom–up	Microwave irradiation	Roasted chickpeas	In the range of 4.5–10.3 nm	No	[240]
Bottom–up	Hydrothermal	Microcrystalline cellulose, hydroxymethylfurfural, and furfural	(6.36 ± 0.54 nm), (5.35 ± 0.56 nm) and (3.94 ± 0.6 nm)	-	[241]
Bottom–up	Microwave irradiation	Citric acid	In the range of 1.5–4.5 nm	Yes	[242]

**Table 7 pharmaceutics-16-00288-t007:** Previous research on the application of CDs in gene therapy.

CDs	Precursors	Synthesis Methods	Size (nm)/Quantum Yield (%)	Cell Line	Ref.
CD-PDMA-PMPD	citric acid	Microwave	50/41.5	COS-7	[264]
Cdots-PEI	tryptophan, Nitrogen, CA	Microwave pyrolysis	4.7 ± 0.8/24.2	MGC-803	[265]
CDs	Citric acid, PEI	Microwave pyrolysis	12–13.2/31.5–48.1	A549	[266]
fc-rPEI-Cdots	Glycerol and PEI	Microwave pyrolysis	143.1/-	H460; 3T3	[267]
CDs	PEI and FA	Hydrothermal	/42	293T; HeLa	[268]
HP-CDs	branched PEI	Hydrothermal	2.25/12.4	HeLa	[245]
CDs/pDNA	Porphyra polysaccharide-EDA	Hydrothermal	<10/56.3	EMSCs	[269]
CDs/pSOX9	Arginine, glucose	Microwave pyrolysis	10–30/12.7	MEFs	[270]
Cationic CDs	Pcyclen, ptaea, citric acid	Hydrothermal	(1.8 ± 0.4), (5.4 ± 2)/-	HeLa	[271]
CDs	Chitosan	Hydrothermal	6–11/-	AGS	[272]

## Data Availability

No new data were created.

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
