# Peer review of "Carbon-Based Nanostructures as Emerging Materials for Gene Delivery Applications"

_pharmaceutics, 2024, doi:10.3390/pharmaceutics16020288_

Round 1
Reviewer 1 Report (Previous Reviewer 2)
Comments and Suggestions for Authors
Dear authors!
Thank you for taking into account all the comments from the previous submission. I think that the review can be accepted for publication.
Author Response
Thank you so much for your valuable comment.

Reviewer 2 Report (Previous Reviewer 3)
Comments and Suggestions for Authors
The authors report an overview of selected carbon-based nanostructures for gene delivery applications. Even if the reported issues may have relevance in nanomedicine, I believe that this manuscript can't be accepted for publication in this Journal since it lacks in novelty (see just as a recent example E. Mostafavi et al., Open Nano, 2022, 7, 100062)
Comments on the Quality of English LanguageMinor editing of English language required
Author Response
The Review that was mentioned [E. Mostafavi et al. Carbon-based nanomaterials in gene therapy. OpenNano 7 (2022) 100062] was fantastic and was incorporated in the manuscript as references 15.
However, The GROSS, OBVIOUS and MAIN difference between Mostafavi’s paper and our review article is that Mostafavi’s paper has mainly and clearly focused on the mechanisms of CELLULAR UPTAKE and transfection of nanocarbon and pDNA complexes into cell lines and vectors in plants and animals. In contrast, our paper focuses on the therapeutic efficacy of gene therapy using carbon-based nanostructure. In addition, we extensively described methods for synthesizing CNTs, CQDs, and NDs taking into account the advantages and disadvantages of each method from the point of view of gene therapy. Therefore, it will certainly fill a gap in the existing literature. Also, the other important carbon-based nanomaterials such as graphene oxide (GO), its reduced form (rGO), and fullerenes have been discussed in the section with this title “ A Perspective on other CBNs” in lines (1238-1377) on page (36-38). Furthermore, in principle, the MAIN and FINAL goal of this review article and probably the novelty was to gather scientific researches related to "the application of carbon-based nanostructures in gene therapy", which gives the reader easier, more complete and comprehensive access to these nanostructures’ advancements. At the same time, it should be kept in mind that a review article does not need to be novel, special and be characterized as a unique innovation.
We would like to respectfully clarify that the nature of this review paper is intended to synthesize existing literature and provide a comprehensive overview rather than introduce a novel subject. We understand the importance of contributing new knowledge in research articles; however, in the context of a review paper, the primary objective is to consolidate and analyze existing information to offer a holistic perspective on the chosen topic. We believe this approach aligns with the intended purpose of review articles in the academic community. If there are specific areas of the paper that you feel could benefit from further clarification or enhancement, We would be more than willing to address those concerns. Your guidance is highly valued, and we are committed to ensuring that the final manuscript meets the expectations of both the journal and its readers.

Reviewer 3 Report (New Reviewer)
Comments and Suggestions for Authors
The manuscript entitled, ‘Carbon based nanostructures as emerging material for gene delivery applications” discussed carbon based systems for gene delivery applications. The article should be modified according to the following points;
1. First of all the abstract needs to be more compact.
2. Initial portions are discussed with CNTs and other carbon nanomaterisls. These are already common and no need of these in this kind of review.
3. The language is very difficult to read. No lucid. Better to rewrite some portions.
4. Author wrote ‘…according to thermodynamic point of view, the formation of the graphitic nanoparticles is the important and essential feature in carbon-dots’ preparation methods…’; what is the thermodynamic discussion here? It is not clear to read also.
5. Some articles would be significance for your references:
(a) Das, P., Ganguly, S., Margel, S., & Gedanken, A. (2021). Tailor made magnetic nanolights: Fabrication to cancer theranostics applications. Nanoscale Advances, 3(24), 6762-6796.
(b) Israel, L. L., Galstyan, A., Holler, E., & Ljubimova, J. Y. (2020). Magnetic iron oxide nanoparticles for imaging, targeting and treatment of primary and metastatic tumors of the brain. Journal of Controlled Release, 320, 45-62.
Comments on the Quality of English LanguageMAJOR
Author Response
The manuscript entitled, ‘Carbon based nanostructures as emerging material for gene delivery applications” discussed carbon based systems for gene delivery applications. The article should be modified according to the following points;
- First of all the abstract needs to be more compact.
Answer: We compacted the Abstract and specified with YELLOW highlight in the revised manuscript.
- Initial portions are discussed with CNTs and other carbon nanomaterisls. These are already common and no need of these in this kind of review.
Answer: Of course, initial portions are discussed with CNTs and other carbon nanomaterisls are common. However, our goal is to provide easier access to the series of information related to nanostructures, from getting to know their structure and properties to their application in gene therapy. For this reason, we will first provide some brief information about the structure, characteristics, synthesis methods and advantages and disadvantages of each one of them, and then we will mention the application of nanostructures in gene therapy. As a result, the reader can get a more general as well as comprehensive view of each nanostructure and its application in the field of gene therapy.
- The language is very difficult to read. No lucid. Better to rewrite some portions.
Answer: Thank you so much. We checked the language, rewrote some parts and modified some sentences.
- Author wrote ‘…according to thermodynamic point of view, the formation of the graphitic nanoparticles is the important and essential feature in carbon-dots’ preparation methods…’; what is the thermodynamic discussion here? It is not clear to read also.
Answer: It means that from a thermodynamic perspective, the formation of graphitic nanoparticles is a crucial and necessary aspect of methods used to prepare carbon dots. This phrase emphasizes that, from a thermodynamic standpoint, the creation of these graphitic nanoparticles is not just a random occurrence but a critical and fundamental aspect of the methods used for preparing carbon dots. In summary, the statement is highlighting the significance of graphitic nanoparticle formation in the context of preparing carbon dots. It suggests that understanding and controlling the thermodynamics of this process are essential for the successful synthesis of carbon dots using various preparation methods. Also, the stability and cohesion of synthesized nanoparticles for biomedical applications are important. In this case, investigating this criterion can be performed through an important thermodynamic parameter called the Flory-Huggins parameter. In other words, the interaction between the components of nanoparticles such as carbon dots is so important. This interaction can be investigated by the Hansen solubility thermodynamic parameter. In fact, the interaction between organic solvents, various saccharides, amino acids, and proteins as a building unit for producing CDs with another component of CDs can be calculated by Hansen solubility thermodynamic parameter. For example, our research team investigated the interaction between SWCNT as a nanocarrier with amphotericin B as a model drug through the thermodynamic parameters mentioned. Also, the stability of synthesized nanocarrier based on SWCNT was studied with the help of the Flory-Huggins parameter [1].
- Some articles would be significance for your references:
(a) Das, P., Ganguly, S., Margel, S., & Gedanken, A. (2021). Tailor made magnetic nanolights: Fabrication to cancer theranostics applications. Nanoscale Advances, 3(24), 6762-6796.
(b) Israel, L. L., Galstyan, A., Holler, E., & Ljubimova, J. Y. (2020). Magnetic iron oxide nanoparticles for imaging, targeting and treatment of primary and metastatic tumors of the brain. Journal of Controlled Release, 320, 45-62.
Answer: Thank you so much for introducing the articles. The above mentioned references are added to the manuscript as references 163 and 11, respectively.

Reviewer 4 Report (New Reviewer)
Comments and Suggestions for Authors
Dear Authors,
Thank you for submitting to MDPI Pharmaceutics. Please see comments below.
Section 2:
--> CNTs: I suggest significantly abbreviating sections 2.1.1 and 2.1.2.
The readers of Pharmaceutics will be more interested in functionalization of CNTs (2.1.3) and applications to gene therapy (2.1.4).
--> That said, both functionalization and applications to gene therapy sections are rather short and, as such, seem a bit incomplete. I suggest significantly abbreviating the CNT intro and synthesis sections: combine into a single section, cut significantly, as the information contained therein has been written about in numerous prior review articles on CNTs as a material.
--> Also, I would expect a section devoted to delivery methods, mechanisms of action, interaction with cells / phagocytosis, transport through the blood stream, etc. Line 129 / 130 deals with this a bit, but only very briefly. The readers of pharmaceutics will be more interested in things like what controls the mechanism of cell ingestion that how the CNTs are made.
--> Also, Section 2 contains no figures. I suggest adding figures from important articles on CNTs used as gene delivery vectors.
-->There should be some assessment of the pros and cons of CNTs as a delivery vector, their likelihood of moving forward for FDA approval, a review of any clinical trials that have occurred using CNTs for gene delivery, an an overview of the toxicity profile / toxicity response.
Section 2.2: Carbon Quantum Dots (CQDs)
--> Again, section on synthesis can be significantly abbreviated for the readership of Pharmaceutics.
--> Section on functionalization, surface chemistry, and interaction with biological surfaces should be significantly expanded.
-->Can authors expand on overall toxicity profile of CQDs? And methods of improving toxicity profiles?
2.3 Nanodiamonds (NDs)
-->As with prior sections, please reduce content on synthesis and expand content on bio-interactions / nano-bio interface / mechanisms of action / cell-NDs physicokinetics.
Section 4 should be significantly expanded.
Overall: Review does a good job of reporting prior findings, but does a poor job of synthesizing those findings into useful, summary information for the reader. Can the authors provide a more constructive, comparative review of the carbon nanomaterials relative to each other, and also relative to other materials / nanomaterials currently being assessed for gene therapy?
Readers of pharmaceutics may also be interested to know what are some common side effects of carbon particle-based gene delivery, what are the common and particle-specific barriers that are keeping these materials from entering clinical trials?
Are any of them currently undergoing clinical trials? Previously?
Readers will also want to know what the body does with these carbon based nanomaterials over long time duration. Does the body break them down? How are they metabolized, if at all? Are the filtered?
Finally, the readership of Pharmaceutics will want to know if any targeting approaches have been successful, as well as if any "stealth" surface chemistry is necessary for hiding from the immune system. The word immune is only mentioned once in the review article. Readers will be very interested in knowing how the immune system responds to such nanomaterials.
English language is good.
Author Response
Dear Authors,
Thank you for submitting to MDPI Pharmaceutics. Please see comments below.
Section 2:
--> CNTs: I suggest significantly abbreviating sections 2.1.1 and 2.1.2.
The readers of Pharmaceutics will be more interested in functionalization of CNTs (2.1.3) and applications to gene therapy (2.1.4).
Answer: we tried to abbreviate sections 2.1.1 and 2.1.2 as much as possible.
--> That said, both functionalization and applications to gene therapy sections are rather short and, as such, seem a bit incomplete. I suggest significantly abbreviating the CNT intro and synthesis sections: combine into a single section, cut significantly, as the information contained therein has been written about in numerous prior review articles on CNTs as a material.
Answer: We design a similar pattern in each section of nanostructures (Below Table). It means that the (1) introduction of nanostructures, (2) Synthesizing methods, (3) Functionalizing Methods, and (4) Application in gene therapy exist in each section of nanostructures (CNTs, CQDs, and NDs) and provide easy access to compare nanostructures with each other.
Abstract |
1- Introduction |
2- Carbon-based Nanostructures |
2.1. Carbon nanotubes (CNTs) 2.1.1. Structure and Properties of CNTs 2.1.2. Synthesis Methods of CNTs 2.1.3. Functionalized CNTs 2.1.4. Application of CNTs in Gene Therapy |
2.2. Carbon Quantum Dots (CQDs) 2-2-1. Structure and Properties of CQDs 2-2-2. Synthesis Methods of CQDs 2-2-2-1. Top-Down Methods 2-2-2-2. Bottom-Up Methods 2-2-3. Functionalized CQD 2-2-4. Application of CQDs in Gene Therapy
|
2-3. Nano-Diamonds (NDs) 2-3-1. Structure and Properties of NDs 2-3-2. Synthesis Methods of NDs 2-3-3. Functionalized NDs 2-3-4. Application of NDs in Gene Therapy |
3. A Perspective On Other Carbon-based Nanostructures |
4. References |
--> Also, I would expect a section devoted to delivery methods, mechanisms of action, interaction with cells / phagocytosis, transport through the blood stream, etc. Line 129 / 130 deals with this a bit, but only very briefly. The readers of pharmaceutics will be more interested in things like what controls the mechanism of cell ingestion that how the CNTs are made.
Answer: All the things (mechnism of action, interaction with cells, transport the blood stream etc) you mentioned exist in “E. Mostafavi et al. Carbon-based nanomaterials in gene therapy. OpenNano 7 (2022) 100062”, that is REF 15. In this article, mechanisms of cellular uptake and transfection of nanocarbon and pDNA complexes into cell lines and vectors in plants and animals were discussed. The mentioned article focused on the mechanisms of cellular uptake and transfection of nanocarbon and pDNA complexes into cell lines and vectors in plants and animals. In contrast, our paper focuses on the therapeutic efficacy of gene therapy using carbon-based nanostructure. In addition, we extensively described methods for synthesizing CNTs, CQDs, and NDs taking into account the advantages and disadvantages of each method from the point of view of gene therapy. Therefore, it will certainly fill a gap in the existing literature. Also, the other important carbon-based nanomaterials such as graphene oxide (GO), its reduced form (rGO), and fullerenes have been discussed in the section with this title “ A Perspective on other CBNs”.
--> Also, Section 2 contains no figures. I suggest adding figures from important articles on CNTs used as gene delivery vectors.
Answer: One Figure was added in section 2.1.4. (Application of CNTs in gene therapy). Figure 2’s explanation specified with YELLOW highlight in this section
-->There should be some assessment of the pros and cons of CNTs as a delivery vector, their likelihood of moving forward for FDA approval, a review of any clinical trials that have occurred using CNTs for gene delivery, and an overview of the toxicity profile / toxicity response.
Answer: In the manuscript, we have included an in-depth analysis of the advantages and disadvantages associated with CNTs as gene delivery vectors. We discuss their unique physicochemical properties that make them promising candidates for efficient gene delivery, along with potential challenges such as cytotoxicity and immunogenicity. By presenting a balanced perspective, readers will gain a more nuanced understanding of the suitability of CNTs in gene therapy. Regarding the potential FDA approval for human use, there are clinical assays on phase I stage that consider the use of CNTs as drug delivery system for the treatment of non-microcytic and advanced lung cancer. Despite this clinical assay is not for gene delivery applications, it holds reasonable promise for future genetic material delivery, with the corresponding optimization process. This information has been included in the resublimed version of the manuscript, at the end of the .1.4. Application of CNTs in Gene Therapy section, in yellow color.
While there have been wide applications of nanodiamonds demonstrated preclinically, clinical translation remains a hurdle but has shown promising early evidence. Clinical studies of nanodiamonds are currently limited to the topical application for root canal therapy (Biomedical applications of nanodiamonds: From drug-delivery to diagnostics. Jingru Xu a,b,c, Edward Kai-Hua Chow).
Despite a lack of widespread FDA approval, CNTs have been studied for decades and plenty of in vivo and in vitro reports have been published (Zare H, Ahmadi S, Ghasemi A, Ghanbari M, Rabiee N, Bagherzadeh M, Karimi M, Webster TJ, Hamblin MR, Mostafavi E. Carbon Nanotubes: Smart Drug/Gene Delivery Carriers. Int J Nanomedicine. 2021 Mar 1;16:1681-1706. doi: 10.2147/IJN.S299448. Erratum in: Int J Nanomedicine. 2021 Oct 28;16:7283-7284. PMID: 33688185; PMCID: PMC7936533.).
Section 2.2: Carbon Quantum Dots (CQDs)
--> Again, section on synthesis can be significantly abbreviated for the readership of Pharmaceutics.
Answer: We tried to abbreviate CQDs sections as much as possible.
--> Section on functionalization, surface chemistry, and interaction with biological surfaces should be significantly expanded.
Answer: We have carefully considered the suggestions made by the reviewer regarding the expansion of the section on functionalization, surface chemistry, and interaction with biological surfaces. While we acknowledge the importance of these aspects in the context of carbon nanostructure-based gene therapy, we believe it is essential to strike a balance between comprehensiveness and readability. In our manuscript, we aimed to provide a comprehensive overview of the carbon nanostructure's role in gene therapy, supported by an extensive bibliography of 368 references. We are concerned that expanding certain sections may lead to an excessive level of complexity and length, potentially compromising the accessibility of the article to a broader readership.
-->Can authors expand on overall toxicity profile of CQDs? And methods of improving toxicity profiles?
Answer: We carefully considered your suggestion to elaborate on the overall toxicity profile of CQDs and the methods employed to improve their toxicity profiles. We acknowledge the importance of addressing this aspect comprehensively to provide a more thorough understanding for the readers. In response to your comment, we would like to highlight that we have indeed addressed the toxicity profile of CQDs in the manuscript. Specifically, we expanded on the overall toxicity profile of CQDs in the revised manuscript, encompassing both in vitro and in vivo studies that contribute to a comprehensive understanding of the potential challenges associated with their use in gene therapy (Lines 814-835). We have dedicated a separate paragraph to discussing strategies for mitigating the cytotoxicity of CQDs, with a focus on the functionality process as one of the key solutions (section 2.2.3.).
2.3 Nanodiamonds (NDs)
-->As with prior sections, please reduce content on synthesis and expand content on bio-interactions / nano-bio interface / mechanisms of action / cell-NDs physicokinetics.
Answer: We tried to abbreviate NDs sections as much as possible and also, we added some contents about the bio-interactions / nano-bio interface / mechanisms of action / cell-NDs physicokinetics that specified with YELLOW highlight in NDs section. (Lines: (955-959),(974-987), and (1216-1224))
Section 4 should be significantly expanded.
Answer: We added one paragraph in section 4 and specified with Yellow highlight, lines 1380-1395.
Overall: Review does a good job of reporting prior findings, but does a poor job of synthesizing those findings into useful, summary information for the reader. Can the authors provide a more constructive, comparative review of the carbon nanomaterials relative to each other, and also relative to other materials / nanomaterials currently being assessed for gene therapy?
Readers of pharmaceutics may also be interested to know what are some common side effects of carbon particle-based gene delivery, what are the common and particle-specific barriers that are keeping these materials from entering clinical trials?
Are any of them currently undergoing clinical trials? Previously?
Readers will also want to know what the body does with these carbon based nanomaterials over long time duration. Does the body break them down? How are they metabolized, if at all? Are the filtered?
Finally, the readership of Pharmaceutics will want to know if any targeting approaches have been successful, as well as if any "stealth" surface chemistry is necessary for hiding from the immune system. The word immune is only mentioned once in the review article. Readers will be very interested in knowing how the immune system responds to such nanomaterials.
Answer: We really appreciate comments and observations of reviewer that we have tried to incorporate in this resubmitted version of the manuscript, in order to enhance the impact for Pharmaceutics journal readers. Regarding the “comparative” point of view, we designed the development of the review, with a general introduction, followed by a similar pattern for the three main carbon based nanostructures (CNTs, CQDs and NDs), which include an introduction section, where structure and general properties of each structure is analyzed. Next section describes the most relevant synthesizing methods; followed by a third section related to the functionalization approaches to enhance their performance, and the last section their application in the gene therapy field. In our opinion, this general structure allows to compare the different carbon nanomaterials in different terms. In any case, and as highly suggested by the reviewer, we have tried to summarize and synthetize this information in the final section (4. Future Prespective on Carbon Nanostructure in Gene Therapy), in yellow color.
We totally agree with reviewer in the sense that still there are some relevant concerns that need to be analyzed and discussed in order to reach the clinical practice of these promising carbon based nanostructures for gene delivery applications. Some of them include the evaluation of side effects, immunogenicity, the relevance of both extracellular and also intracellular biological barriers that genetic material needs to cross, and strategies for overcoming such barriers, or their metabolism in human body once administered, as rightly highlighted by the reviewer. However, although highly interesting, these concerns initially scape from the main goal of the review, and we have not discussed them in depth, although with them there are many references along all the manuscript, that can reefer the Pharmaceuticals readers to such relevant information. Please, note that we have reviewed three main types of carbon-based nanostructures that can be used in gene delivery applications for preclinical purposes, such as CNTs, CQDs, and NDs. In addition, other similar carbon-based nanostructures such as GO and FU were considered. For each class of nanostructures the following were explored in depth: the structures and properties, the synthesis methods and functionalization approaches, and the applications in gene therapy. At the end, the review manuscript contains 7 tables, 7 figures and a total of 357 references that cover the oldest article dates back to 1972 and the most recent one to 2023. Therefore, adding other relevant information, although really interesting would increase in excess the content and length of such review.

Reviewer 5 Report (New Reviewer)
Comments and Suggestions for Authors
The manuscript entitled “Carbon based nanostructures as emerging material for gene delivery applications” reports a substantial review on three particular types of carbon-based nanostructures that can be used in gene delivery applications.
In particular, carbon nanotubes (CNTs), carbon quantum dots (CQDs), nanodiamonds (NDs) and other similar carbon-based nanostructures were considered.
For each class of nanostructures the following were explored in depth: the structures and properties, the synthesis methods, the applications in gene therapy.
The article is developed in a systematic and well-organized way. It is clear and easy to read.
The review considered 357 articles covering a wide time period; in fact, the oldest article dates back to 1972 and the most recent one to 2023.
The topic covered is interesting and consistent with the purpose of the newspaper. In general my opinion is overall positive.
I have no particular suggestions to send to the authors, I believe that the manuscript is, already in the present form, to be considered for publication.
Author Response
Thank you so much for your valuable comments.
Round 2
Reviewer 2 Report (Previous Reviewer 3)
Comments and Suggestions for Authors
The authors replied to almost all the comments. I believe that this revised version of the manuscript can be accepted for publication in this Journal.
Comments on the Quality of English LanguageMinor editing of English language required
Reviewer 3 Report (New Reviewer)
Comments and Suggestions for Authors
This can be published in its present form.
Reviewer 4 Report (New Reviewer)
Comments and Suggestions for Authors
Dear Authors -- Thank you for providing a complete and timely response. Your updates are satisfactory and I believe improve the readability and usefulness of the paper overall. Thank you and all the best for your future endeavors.
This manuscript is a resubmission of an earlier submission. The following is a list of the peer review reports and author responses from that submission.
Round 1
Reviewer 1 Report
Comments and Suggestions for Authors
The manuscript reviews the state of art on the application of carbon nanomaterials as non-viral carriers for gene delivery. Noteworthy is the very broad coverage of the topic - from carbon nanotubes through quantum dots to nanodiamond. Previous review works [S. Taghavari et al. Hybrid carbon-based materials for gene delivery in cancer therapy. Journal of Controlled Release 318 (2020) 158; E. Mostafavi et al. Carbon-based nanomaterials in gene therapy. OpenNano 7 (2022) 100062] have mainly focused on the mechanisms of cellular uptake and transfection of nanocarbon and pDNA complexes into cell lines. In contrast, this paper focuses on the therapeutic efficacy of gene therapy using nanocarbon. The authors also extensively described methods for synthesizing nanotubes, carbon quantum dots and nanodiamonds taking into account the advantages and disadvantages of each method from the point of view gene therapy. Therefore, it will certainly fill a gap in the existing literature. I recommend publication of the paper in the journal Pharmaceutics after taking into account minor editorial corrections.
Comments on the Quality of English LanguageEditorial corrections:
Line 144: “chemical vapor deposition” instead of “Chemical vapor deposition”.
Line 174: „50 Â μm” ???
Line 227: “650 t0 850 °C” instead of „650 t0 850 °C”.
Line 441: “PDs” - the development of this abbreviation is expected.
Author Response
Thank you so much for your comments. The two introduced papers were fantastic and have been entered in the manuscript as Refs 13 and 14.
Editorial corrections:
Line 144: “chemical vapor deposition” instead of “Chemical vapor deposition”.
Answer: It was done and highlighted in YELLOW color. (Line 147)
Line 174: „50 Â μm” ???
Answer: A micrometre (American spelling: micrometer: symbol  μm) is an SI unit length equal to one millionth of a meter, or equivalently, one thousandth of a millimeter. It is also commonly known as a micron. The symbol  μm character Âμ (Unicode character U+00B5; HTML micro) is the “micro sign”, which should look identical to the Greek letter mu (I ¼). Micron is an abbreviation of micrometer and  μm is more of a symbol of micrometer.
Line 227: “650 t0 850 °C” instead of „650 t0 850 °C”.
Answer: It was done and highlighted in YELLOW color. (Line 230)
Line 441: “PDs” - the development of this abbreviation is expected.
Answer: “PDs” is the abbreviation of polymer dots. We introduce it in line 390.

Reviewer 2 Report
Comments and Suggestions for Authors
The review by Dr. Mehrdad Mozaffarian et. al. is devoted to the use of carbon based nanostructures as gene therapy. The review is well structured and written, however, there are several things to discuss. My comments:
1. Line 431-432. "In recent years, CQD preparation has been dependent on carbon sources and reaction methods." - can you please explain this phrase? The CQD never have been depended on carbon sources and reaction methods and only in recent years it became true? Nonsense
2. Figure 2 seem to mislead the reader - it looks like laser ablation method (top-down) uses molecular precursors, as well as in hydrothermal method (bottom-up), macroscopic carbon structures are usually used.
3. For CNTs authors also included information about its functionalization - can authors please add the same paragraph for CQD and nanodiamonds?
4. According to the journal rules (https://www.mdpi.com/journal/pharmaceutics/instructions):
Reviews offer a comprehensive analysis of the existing literature within a field of study, identifying current gaps or problems. They should be critical and constructive and provide recommendations for future research. No new, unpublished data should be presented. The structure can include an Abstract, Keywords, Introduction, Relevant Sections, Discussion, Conclusions, and Future Directions, with a suggested minimum word count of 4000 words.
However, it is not completely clear, what are current gaps or problem(s)? As well as future directions are not mentioned.
Author Response
The review by Dr. Mehrdad Mozaffarian et. al. is devoted to the use of carbon based nanostructures as gene therapy. The review is well structured and written; however, there are several things to discuss. My comments:
- Line 431-432. "In recent years, CQD preparation has been dependent on carbon sources and reaction methods." - can you please explain this phrase? The CQD never have been depended on carbon sources and reaction methods and only in recent years, it became true? Nonsense
Answer: It means that “Carbon sources and reaction methods are two important parameters in choosing the methods of synthesizing CQDs”. As we will explain in the following sentence, “CQDs’ synthesizing methods can be classified into two groups: top-down and bottom-up”
- Figure 2 seem to mislead the reader - it looks like laser ablation method (top-down) uses molecular precursors, as well as in hydrothermal method (bottom-up), macroscopic carbon structures are usually used.
Answer: Carbon sources in Bottom-Up methods are macroscopic carbon structures such as fibers and films made from graphene or CNTs. However, Carbon sources in Top-Down methods are molecular precursors such as sucrose, citric acid, ascorbic acid, sucrose-citric acid, sucrose- ascorbic acid and they all exhibit strong photoluminescence in the visible region. The mechanism for synthesis of carbon dots from sucrose/ascorbic acid/citric acid involves their carbonization.
- For CNTs authors also included information about its functionalization - can authors please add the same paragraph for CQD and nanodiamonds?
Answer: The functionalization of CQDs and NDs are very similar and the same as the methods of functionalizing CNTs. For this reason, in order to avoid repetition and offer duplicate information, we did not mention their functionalization in the CQDs and NDs section, because the discussion would have been long and repetitive.
- According to the journal rules (https://www.mdpi.com/journal/pharmaceutics/instructions): Reviews offer a comprehensive analysis of the existing literature within a field of study, identifying current gaps or problems. They should be critical and constructive and provide recommendations for future research. No new, unpublished data should be presented. The structure can include an Abstract, Keywords, Introduction, Relevant Sections, Discussion, Conclusions, and Future Directions, with a suggested minimum word count of 4000 words.
However, it is not completely clear, what are current gaps or problem(s)? As well as future directions are not mentioned.
Answer: Challenges and future directions are explained in LINE 1072 to 1092, as well as mentioned below:
Future Perspective on Carbon Nanostructure in Gene Therapy
Researchers all over the world believe that developments, and profiling of different non-viral nanostructures for gene delivery would have a wonderful and great impact on future gene therapies. In the future, we will observe some of the required high-cost treatments for targeted diseases replaced with inexpensive therapies. New genetic treatments will be created for both ordinary as well as uncommon diseases. Researchers have reported that similar to other biologics, gene therapies are also expected to see notable signs of progress in the coming years.
Nanostructure-based gene delivery systems are emerging as an exciting new method with the possibility to overcome known obstacles and multiple biological and medical requirements. The main characteristic of all delivery systems is the need for transfer of genetic materials into the cells, where their favorable operations will be exposed. Therefore, a requirement for a delivery system is not to let the genetic materials to enter the cells passively. The emerging field of nano-gene delivery introduces the utilization of novel nanostructures and their features to generate vectors that can effectively deliver different genetic material into a variety of various types of cells. The physicochemical properties of the carbon-based delivery vectors can be applied to address the ongoing challenges existing in delivering the genetic materials at in vitro and in vivo scales. While there is an increasing interest in nanostructure-based gene delivery systems, the string is still in the initial steps, and there is a powerful need to figure out nanostructures and their physicochemical features on biological grounds.

Reviewer 3 Report
Comments and Suggestions for Authors
The authors report an overview of carbon-based nanostructures nanoplatforms for gene delivery systems. Even if the reported issues may have relevance in pharmaceutical field, I believe that this manuscript can’t be accepted for publication in this Journal, since it mainly lacks in novelty. Other authors have reported the recent advances in the use of Carbon-based nanomaterials in gene therapy (see just as as example, E. Mostafavi and H, Zare, OpenNano, 2022, 7, 100062 or S. Taghavi et al., J. Control. Release, 2020, 318, 58‒175)
Moreover, why did the authors focus their attention only on carbon nanotubes, carbon quantum dots and nanodiamonds? Other important carbon based nanomaterials such as graphene oxide (GO), its reduced form (rGO) and fullerenes have been successfully employed as gene delivery systems. In my opinion, there is no reason why these carbon-based nanomaterials should be excluded.
Comments on the Quality of English LanguageMinor editing of English language required.
Author Response
The authors report an overview of carbon-based nanostructures nanoplatforms for gene delivery systems. Even if the reported issues may have relevance in pharmaceutical field, I believe that this manuscript can’t be accepted for publication in this Journal, since it mainly lacks in novelty. Other authors have reported the recent advances in the use of Carbon-based nanomaterials in gene therapy (see just as as example, E. Mostafavi and H, Zare, OpenNano, 2022, 7, 100062 or S. Taghavi et al., J. Control. Release, 2020, 318, 58‒175).
Moreover, why did the authors focus their attention only on carbon nanotubes, carbon quantum dots and nanodiamonds? Other important carbon based nanomaterials such as graphene oxide (GO), its reduced form (rGO) and fullerenes have been successfully employed as gene delivery systems. In my opinion, there is no reason why these carbon-based nanomaterials should be excluded.
Answer: Two reviews that were mentioned [S. Taghavari et al. Hybrid carbon-based materials for gene delivery in cancer therapy. Journal of Controlled Release 318 (2020) 158; E. Mostafavi et al. Carbon-based nanomaterials in gene therapy. OpenNano 7 (2022) 100062] were fantastic and were entered into the manuscript as references 13 and 14. However, an according to the opinion of reviewer 1, these two papers have mainly focused on the mechanisms of cellular uptake and transfection of nanocarbon and pDNA complexes into cell lines. In contrast, our paper focuses on the therapeutic efficacy of gene therapy using carbon-based nanostructure. In addition, we extensively described methods for synthesizing CNTs, CQDs, and NDs taking into account the advantages and disadvantages of each method from the point of view of gene therapy. Therefore, it will certainly fill a gap in the existing literature.
We agree with the reviewer in the sense that other important carbon-based nanomaterials such as graphene oxide (GO), its reduced form (rGO), and also fullerenes have been successfully employed as gene delivery systems. In any case, in this manuscript, we have focused the interest only in 3 dimensional carbon based nanostructures such as CNTs, CQDs and NDs that are familiar for us, since we have wide experience working with them [1-5]. Additionally, we have highlighted in the manuscript, graphene is also present in CNTs (rolled layers of graphene) and also, in some kind of CQDs (we have mentioned graphene oxide quantum dots in lines 568 and 572). Consequently, although other similar nanostructures such as GO, rGO and fullerenes have shown promising results as efficient and safe genetic material delivery systems, we have not included them in this manuscript to avoid a long and extensive manuscript. Instead, we have referred the readers of the resubmitted version of the manuscript to recent references that deal extensively with this issue, in the page 2 line 73-75.
Comments on the Quality of English Language
Minor editing of English language required.
Answer: Corrected as much as possible.
- Al Qtaish, N.; Gallego, I.; Paredes, A.J.; Villate-Beitia, I.; Soto-Sánchez, C.; Martínez-Navarrete, G.; Sainz-Ramos, M.; Lopez-Mendez, T.B.; Fernández, E.; Puras, G. Nanodiamond Integration into Niosomes as an Emerging and Efficient Gene Therapy Nanoplatform for Central Nervous System Diseases. ACS applied materials & interfaces 2022, 14, 13665-13677.
- Yazdani, S.; Mozaffarian, M.; Pazuki, G.; Hadidi, N. Application of Flory-Huggins Model in Experimental and Theoretical Study of Stability of Amphotericin B on Nanocarrier Based on Functionalized Carbon Nanotube. Journal of Molecular Liquids 2022, 119519.
- Yazdani, S.; Mozaffarian, M.; Pazuki, G.; Hadidi, N.; Gallego, I.; Puras, G.; Pedraz, J.L. Design of double functionalized carbon nanotube for amphotericin B and genetic material delivery. Scientific Reports 2022, 12, 1-15.
- Hadidi, N.; Shahbahrami Moghadam, N.; Pazuki, G.; Parvin, P.; Shahi, F. In Vitro Evaluation of DSPE-PEG (5000) Amine SWCNT Toxicity and Efficacy as a Novel Nanovector Candidate in Photothermal Therapy by Response Surface Methodology (RSM). Cells 2021, 10, 2874.
- Hadidi, N.; Mohebbi, M. Anti-Infective and Toxicity Properties of Carbon Based Materials: Graphene and Functionalized Carbon Nanotubes. Microorganisms 2022, 10, 2439.

Round 2
Reviewer 2 Report
Comments and Suggestions for Authors
Dear Dr. Mehrdad Mozaffarian et. al. thank you for your revised manuscript, however, I did not saw any key changes.
Line 433-433. "In recent years, CQD preparation has been dependent on carbon sources and reaction methods." is the same. If, as you wrote, it means that “Carbon sources and reaction methods are two important parameters in choosing the methods of synthesizing CQDs” - so why not to white it in the text instead of the first phrase?
The reviewer softly insists on the addition of the paragraphs about functionalization of CQD and nanodiamonds. It is not the same as for CNT.
Reviewer 3 Report
Comments and Suggestions for Authors
I still believe that the issues reported in this review lack in novelty and that no important contribution has been maded to justify the publication of this review. Furthermore, other important carbon-based nanomaterials have not been reviewed.
Comments on the Quality of English LanguageMinor editing of English language required.